# Revealing the role of double-layer micro-environments in pH-dependent oxygen reduction activity over metal-nitrogen-carbon catalysts

Peng Li[1,3], Yuzhou Jiao[1,3], Yaner Ruan[2], Houguo Fei[1], Yana Men[1], Cunlan Guo[1], Yuen Wu [2] ✉ & Shengli Chen [1] ✉

A standing puzzle in electrochemistry is that why the metal-nitrogen-carbon catalysts generally exhibit dramatic activity drop for oxygen reduction when traversing from alkaline to acid. Here, taking FeCo-N$_6$-C double-atom catalyst as a model system and combining the ab initio molecular dynamics simulation and in situ surface-enhanced infrared absorption spectroscopy, we show that it is the significantly distinct interfacial double-layer structures, rather than the energetics of multiple reaction steps, that cause the pH-dependent oxygen reduction activity on metal-nitrogen-carbon catalysts. Specifically, the greatly disparate charge densities on electrode surfaces render different orientations of interfacial water under alkaline and acid oxygen reduction conditions, thereby affecting the formation of hydrogen bonds between the surface oxygenated intermediates and the interfacial water molecules, eventually controlling the kinetics of the proton-coupled electron transfer steps. The present findings may open new and feasible avenues for the design of advanced metal-nitrogen-carbon catalysts for proton exchange membrane fuel cells.

Metal-nitrogen-carbon (M-N-C) catalysts, mainly including single-atom catalysts (SACs) and double-atom catalysts (DACs), are one of the most promising alternatives to costly Pt-based materials for oxygen reduction reaction (ORR)[1–3], which is the cornerstone of various important energy conversion technologies, such as fuel cells and metal-air batteries[4–6]. In recent years, tremendous efforts have been devoted to improving the ORR performance, electrocatalytic stability and selectivity of M-N-C catalysts by modulating the local coordination environment of metal center, adjusting the electronic structures through doping of light heteroatoms and so on[2,7]. Although great progress in material development, the ORR mechanism on M-N-C catalysts is still elusive and clouded by several long-standing fundamental puzzles. For

example, numerous studies reported that the M-N-C catalysts usually exhibit superior ORR activity in alkaline electrolytes; while in acid electrolytes, the ORR performance is fairly inferior[8–10]. Unfortunately, the mechanistic origin of such a dramatic ORR activity gap in alkaline and acid medias remains an unknown "black box", which has not only constituted a major puzzle in modern electrochemistry but seriously retarded the further development of proton exchange membrane fuel cells by applying the low-cost M-N-C catalysts.

So far, only few studies have attempted to interpret the huge pH-dependence of ORR performance on M-N-C catalysts. Jaouen et al. suggested that the significant degradation of ORR electrocatalytic activity in acid media can be attributed to the protonation of surface

[1]Hubei Key Laboratory of Electrochemical Power Sources, College of Chemistry and Molecular Sciences, Wuhan University, Wuhan 430072, China. [2]School of Chemistry and Materials Science, Collaborative Innovation Center of Chemistry for Energy Materials (iChEM), University of Science and Technology of China, Hefei 230026, China. [3]These authors contributed equally: Peng Li, Yuzhou Jiao. ✉e-mail: yuenwu@ustc.edu.cn; slchen@whu.edu.cn

nitrogen groups and the subsequent anion binding[11,12]. Mukerjee et al. stated that it is the difference of electron transfer mechanisms under alkaline and acid ORR conditions that accounts for the activity gap[8,13]. Concretely, in alkaline media, the specifically adsorbed hydroxyl species can promote an inner-sphere electron transfer ORR mechanism that mainly facilitates the four-electron pathway; while in acid media, the ORR undergoes the outer-sphere electron transfer mechanism that mainly contributes to the two-electron $H_2O_2$ formation. Recently, Li et al. pointed out that during the ORR, the Fe sites of Fe-N-C catalysts are occupied by the *OH intermediate to form a $FeN_4$-OH center at acid pH, while covered by the *O intermediate to form a $FeN_4$-O center at alkaline pH, thereby attributing the activity difference of Fe-N-C to the different intermediate occupancy on Fe atom and the related electronic structure regulation of Fe site exerted by these occupying species[14]. Similarly, Zhou et al. has also revealed that the in situ active site transformation to *O-$FeN_4$ and *OH-$FeN_4$ could the origin of the higher activity of Fe-N-C in alkaline media[15]. However, it is certainly fair to say that the understanding of the dramatic ORR activity gap between alkaline and acid medias is still incomplete and far from satisfactory. For instance, the pH-dependent interfacial environment and its critical effects in the electrocatalytic reaction mechanism, thermodynamics and kinetics are ignored, especially on the atomic-molecular scale[16–19].

Recently, DACs have received ever-increasing attention for ORR profiting from the synergistic effect between adjacent metal active sites[20–23]. Therefore, taking FeCo-$N_6$-C catalyst as a model system herein, the mechanistic origin of the pH-dependent ORR activity has been comprehensively investigated by combining the ab initio molecular dynamics (AIMD) simulation, the slow-growth approach and in situ surface-enhanced infrared absorption spectroscopy (SEIRAS) with the attenuated total reflection (ATR) configuration. It is found that at ORR potentials, the FeCo-$N_6$-C electrode is almost uncharged at alkaline interface while much positively charged at acid interface, which thus leads to the dramatic discrepancy of interfacial double-layer microenvironment around Fe-Co reactive center. Specifically, the interfacial water molecules at alkaline interface are orientated disorderly and thus can naturally form the hydrogen bonds with the surface oxygenated intermediates, which is fairly vital for the proton-coupled electron transfer (PCET) steps during the ORR process. Per contra, all interfacial water molecules around metal centers are orderly arranged in the form of O-down configurations at acid interface due to the strong electric filed polarization effect, which seriously breaks the hydrogen bonds between the oxygenated intermediates and the interfacial water solvent, thereby hinders the proton transfer in PCET steps, and eventually brings about the sluggish ORR reaction kinetics. In addition, we reveal that the great activity drop from alkaline to acid cannot be explained by the difference of reaction energetics along the ORR pathways, because the potential-determining step (PDS) of ORR at alkaline interface possess a higher reaction free energy than that at acid interface, which unequivocally substantiates that the pH-dependent ORR activity on FeCo-$N_6$-C catalyst is attributed to the distinct PCET reaction kinetics factors caused by the interfacial water orientation and hydrogen bonds. Our work not only provides a unique molecular-level insight to understand the huge ORR activity gap of M-N-C catalysts at different pH values, but also highlights the fundamental and technological significance of electrochemical interface structure for the further development of modern electrochemistry science and electrochemical energy conversion and storage.

## Results and discussion

### Double-layer microenvironments under alkaline and acid ORR conditions

In order to obtain the molecular pictures of double-layer microenvironments around metal reactive center at alkaline and acid interfaces under ORR conditions, the FeCo-$N_6$-C/electrolyte interfaces with adsorbed $O_2$ molecules on the Fe-Co bridge sites have been simulated through AIMD (see Methods and Supplementary Figs. 1 and 2 for details of the computational simulation and model construction). Figure 1 shows that one $K^+$ cation and one $OH^-$ anion are introduced into the water film to simulate the alkaline environment and meantime to drive the electrode potential to -1.10 V (unless stated, electrode potentials in this work are referenced to the reversible hydrogen electrode, RHE) for pH = 13 (Fig. 1a); while at acid interface, one $H_3O^+$ cation and two $F^-$ anions are required to adjust the electrode potential to -0.88 V for pH = 1 (Fig. 1b). It is worth noting that the simulated potentials in alkaline and acid systems are in line with the

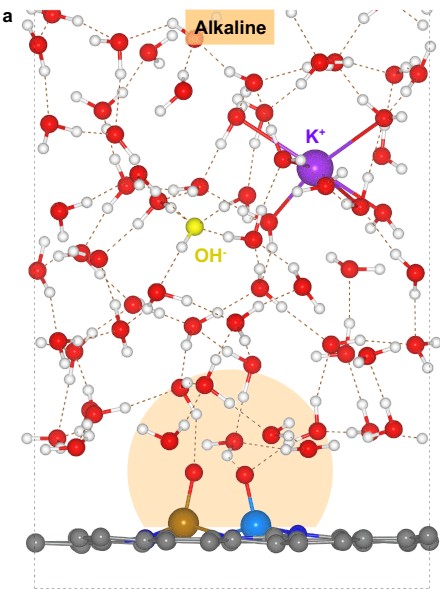
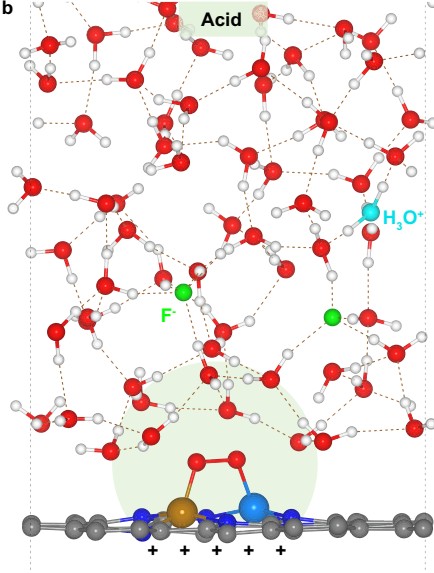

**Fig. 1 | AIMD simulated interfacial structures.** Representative snapshots of the interfacial structures at ORR potentials on the $O_2$ adsorbed FeCo-$N_6$-C electrode surfaces for alkaline (**a**) and acid (**b**) systems. The arched shadows represent the double-layer microenvironments around reactive centers. The Fe, Co, N, C, O, H, $K^+$, $OH^-$, $H_3O^+$ and $F^-$ are colored with brown, sky blue, blue, gray, red, white, purple, yellow, cyan and green, respectively (similarly hereinafter). The brown dashed lines represent the hydrogen bonds.

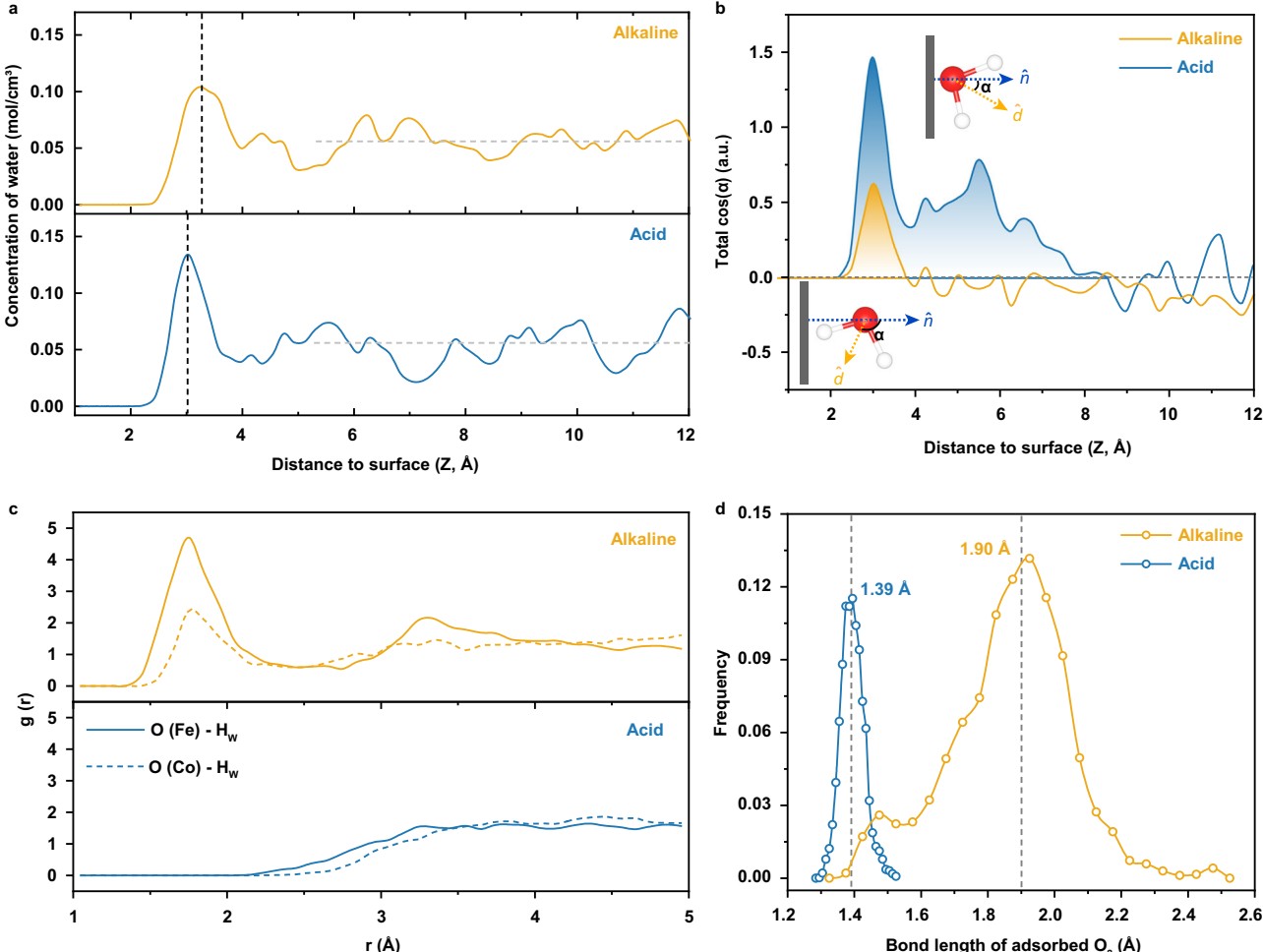

**Fig. 2 | Double-layer microenvironments at alkaline and acid interfaces.**
**a** Distribution profiles of water concentration along the surface normal direction at alkaline and acid interfaces. The positions of water molecules are indicated by their oxygen atoms, and the zero in the z-coordinate represent the position of $N_6$-C plane. The horizontal gray dashed lines represent the bulk water concentration (0.056 mol/cm³). **b** Distribution profiles of water dipole orientation along the surface normal direction. The insets show that α is defined as the angle between the vector of water dipole ($\hat{d}$) and the surface normal ($\hat{n}$). **c** Radial distribution functions between the O atoms of adsorbed $O_2$ molecule and the H atoms of interfacial water. **d** Statistical distributions of the O-O bond length of adsorbed $O_2$ at alkaline and acid interfaces.

corresponding experimental values at which ORR takes place in 0.1 M KOH and 0.1 M $HClO_4$ electrolytes, respectively[23,24]. More importantly, this indicates that under the ORR conditions, the FeCo-$N_6$-C electrode is almost uncharged at alkaline interface while more positively charged at acid interface. It can be expected that the difference in the surface charge density will lead to significant discrepancy of interfacial double-layer microenvironments in alkaline and acid medias.

To accurately capture the difference between the double-layer structures at alkaline and acid interfaces, the statistics of concentration distributions of water molecules along the surface normal direction are first performed. Figure 2a shows that there exist distinct water concentration peaks at both alkaline and acid interfaces, which locates at ~3.29 Å and ~3.03 Å away from the FeCo-$N_6$-C electrode surface, respectively. The smaller distance at acid interface can be attributed to the strong electrostatic attraction exerted by the positively charged electrode on interfacial water molecules. In addition, it can be noted that the interfacial water concentration peak is much sharper in acid system than that in alkaline system, which may be related to the difference in the order degrees of interfacial water arrangement. To verity this hypothesis, the water dipole orientations at alkaline and acid interfaces have been analyzed by counting the total cosα distribution profiles along the AIMD trajectories, where α is the angle between the

vector of water dipole ($\hat{d}$) and the surface normal ($\hat{n}$), as depicted in Fig. 2b. It is apparent that at acid interface, the distribution of total cosα displays a series of peaks with much higher amplitudes within ~7.7 Å from the electrode surface, which implies that the interfacial water molecules are inclined to be orderly arranged in the form of O-down configurations due to the positively charged electrode surface. By contrast, there only exists a fairly weak and narrow peak at alkaline interface, indicating that the interfacial water bears little net orientation dipole, namely is orientated disorderly. As a consequent, it is imaginable that the O atoms of adsorbed $O_2$ can naturally form the hydrogen bonds with the H atoms of interfacial water in alkaline media, which can be verified by the radial distribution functions that exhibit sharp peaks around 1.76 Å (Fig. 2c). On the contrary, such hydrogen bonds are missing at acid interface (Figs. 1 and 2c), because there are no interfacial water molecules with H atoms pointing to the FeCo-$N_6$-C electrode surface.

Furthermore, the O-O bond lengths of adsorbed $O_2$ molecules along the AIMD trajectories have also been extracted (Supplementary Fig. 3). It is worth emphasizing that the statistical averages of the O-O bond lengths of $O_2$ molecules adsorbed on Fe-Co bridge sites at alkaline and acid interfaces are ~1.90 Å and ~1.39 Å, respectively (Fig. 2d), which means that the adsorbed $O_2$ molecule at alkaline

interface tends to spontaneously dissociates into two *O intermediates due to the assistance of hydrogen bonds formed with interfacial water. On the other hand, it should be noted that there exist no hydronium ions in the electric double layer at acid interface (Fig. 1b), because the electrode surface is much positively charged. Therefore, at both acid and alkaline interfaces, it will be the water molecules that serve as the proton donors for the multistep PCET reactions in ORR process. Given the fact that the water molecules are orderly arranged in O-down configuration at acid interface, it can be easily expected that the PCET reactions in acid ORR process are going to be confronted with much higher kinetic barriers than that in alkaline ORR system, due to that the interfacial water must overcome the strong electric field to flip into the H-down configuration and thereby provide the H atom to oxygenated intermediates. In addition, it is worth mentioning that such scenarios are also likely to exist for SACs. As shown in Supplementary Fig. 4, it is clear that both potentials of zero total charge (PZTCs) of the Fe-$N_4$-C/water and Co-$N_4$-C/water interfaces with *$O_2$ are distinctly lower than the ORR potential at pH = 1 while fairly close to the ORR potential at pH = 13, implying that the SAC electrodes are also much positively charged in acid and almost uncharged in alkaline.

Furthermore, the influence of active site density (SD) of FeCo-$N_6$-C catalyst on the double-layer microenvironments at alkaline and acid ORR interfaces is evaluated, because the SD of the model shown in Fig. 1 ($9.2 \times 10^{20}$ site/g) is much higher than the experimental SD value of the as-prepared FeCo-$N_6$-C sample (~$1.1 \times 10^{20}$ site/g)[20]. As shown in Supplementary Figs. 5 and 6, two enlarged FeCo-$N_6$-C/water interface models, which possess the SD of $4.6 \times 10^{20}$ site/g and $2.3 \times 10^{20}$ site/g, respectively, are established and simulated to obtain the potentials of zero charge (PZCs). Supplementary Fig. 6e shows that for the clean FeCo-$N_6$-C/water interfaces, the potential of zero free charge (PZFC) exhibits obvious decrease with the SD decreasing, but not monotonically. By contrast, the PZTC of $O_2$ adsorbed FeCo-$N_6$-C/water interface decreases monotonically with the SD decreasing. Such change trends of the PZTC and PZFC for FeCo-$N_6$-C/water interfaces indicate that the SD in model indeed affects the magnitude of the surface dipole potential induced by intermediate and thus the calculated values of the electrode potentials. However, it should be noted that with the SD decreasing from $9.2 \times 10^{20}$ site/g to $2.3 \times 10^{20}$ site/g which is similar to the experimental SD, the decrease in PZTC and PZFC of FeCo-$N_6$-C/water interface does not change the significant difference of surface charge densities on FeCo-$N_6$-C electrodes in alkaline and acid medias, and thereby the significant difference of the double-layer microenvironments at alkaline and acid ORR interfaces. Specifically, as shown in Supplementary Fig. 6f, when the SD decreases, the PZFC (horizontal dashed line) and the PZTC (horizontal solid line) of FeCo-$N_6$-C will gradually shift downward, consequently leading to the further deviation (marked by red arrows) of them relative to the ORR reaction potential at both acid and alkaline pHs. Therefore, the electrode will always possess much higher positively surface charge density in acidic electrolyte compared to that in alkaline electrolyte, indicating that the water molecules at acid interface will be much more orderly and orientally-rigid. Consequently, one can surmise that the distinct discrepancy of the double-layer microenvironments around reactive centers may be the root of the dramatic ORR activity gap in alkaline and acid electrolytes for M-N-C catalysts.

### In situ SEIRAS verification of the double-layer microenvironment difference

To experimentally rationalize the AIMD-simulated double-layer microenvironments in alkaline and acid electrolytes and their difference, the in situ SEIRAS measurements are performed to probe the interfacial water structures. Figure 3a, b are the in situ SEIRAS spectra of ORR on FeCo-$N_6$-C catalyst recorded at potentials from 0.8 V to 0.4 V vs RHE in $O_2$-saturated solutions of 0.1 M KOH and 0.1 M HClO$_4$, which clearly show the O-H stretching mode (~2800–3600 cm$^{-1}$) and

H-O-H bending mode (~1600–1700 cm$^{-1}$) of interfacial water. It is noted that both the frequencies and intensities of these two peaks exhibit an obvious potential dependence, suggesting that the obtained spectral signals are primarily derived from the first few water layers close to the electrode surface[25]. More importantly, the O-H stretching peaks at various potentials in alkaline electrolytes mainly locate around ~3200 cm$^{-1}$, which is redshifted by 200 cm$^{-1}$ compared to that in acid media (~3400 cm$^{-1}$). This implies that the interfacial water molecules in alkaline environment are strongly hydrogen-bonded due to their disordered orientation on an almost uncharged electrode surface and similarity to the bulk water. By contrast, the interfacial water molecules in acid electrolyte are orderly arranged in O-down configurations on a much positively charged electrode surface, which brings about the decrease in the number of hydrogen bonds of interfacial water molecules and thus higher O-H stretching vibration frequency.

Moreover, the O-H stretching peaks are further deconvoluted into three distinct components through Gaussian fitting, as shown in Fig. 3c for 0.7 V and Supplementary Fig. 7 for other potentials, corresponding to three types of O-H stretching vibrations. Thereinto, the components locating at ~3400 cm$^{-1}$ (green shadow) and ~3200 cm$^{-1}$ (purple shadow) exist in both alkaline and acid electrolytes and are assigned to the 4-coordinated hydrogen-bonded water (4-HB-$H_2O$) and 2-coordinated hydrogen-bonded water (2-HB-$H_2O$), respectively[26–31]. Figure 3d, e show that the relative proportion of 4-HB-$H_2O$ in alkaline media (~59%) is much higher than that in acid electrolyte (~32%), while the relative proportion of 2-HB-$H_2O$ is opposite (~16% vs ~54%), which imply that the interfacial water molecules are indeed with stronger hydrogen-bond interactions in alkaline system. Besides, it is found that there also exists another component possessing a fairly low frequency of ~2950 cm$^{-1}$ (yellow shadow in Fig. 3c) and an average relative proportion of ~25% (upper panel in Fig. 3f) in alkaline system, which may correspond to the interfacial water molecules that forms hydrogen-bonds with the oxygenated intermediates (termed as intermediate-bound $H_2O$). To support this conjecture, the computational vibrational density of states (VDOS) of intermediate-bound $H_2O$ at the simulated alkaline interface is calculated. Supplementary Fig. 8 shows that the VDOS of intermediate-bound $H_2O$ does contribute obvious vibration peaks located from 2800 cm$^{-1}$ to 3000 cm$^{-1}$, and meanwhile the strongly intermediate-bound water molecules with five hydrogen bonds can be directly observed along the AIMD trajectory (inset of Supplementary Fig. 8), which confirms the accuracy of the above assignment for the low frequency component and provides a solid support for the alkaline double-layer microenvironment. Conversely, the third component in acid electrolyte possesses a quite high frequency of ~3550 cm$^{-1}$ (blue shadow in Fig. 3c) and an average relative proportion of only ~14% (lower panel in Fig. 3f), which is associated with the isolated $H_2O$ that may form due to the further destroy of hydrogen-bond interaction exerted by the ions in double-layer[26,29]. To sum up, it can be seen that the salient consistency between the experimental spectroscopy, computational spectroscopy and AIMD simulation results could unequivocally substantiate the discrepancy of the double-layer microenvironments at alkaline and acid ORR interfaces. Certainly, apart from the characterization of the interfacial water structure by in situ spectroscopy, the direct measurement and/or indirect evaluation of surface charge density are also worth exploring to provide additional experimental verification for the simulated results, but this is still challenging so far.

### Determination of ORR pathways at alkaline and acid interfaces

Based on the AIMD-simulated and SEIRAS-confirmed interfaces, the complete ORR pathways and reaction energetics in alkaline and acid medias were further explored. Starting with the $O_2$ adsorption (Fig. 1), all possible reaction scenarios in each subsequent elementary step have been considered, and the one with the lowest energy is selected as the reaction product (Supplementary Fig. 9). Figure 4a, b shows the

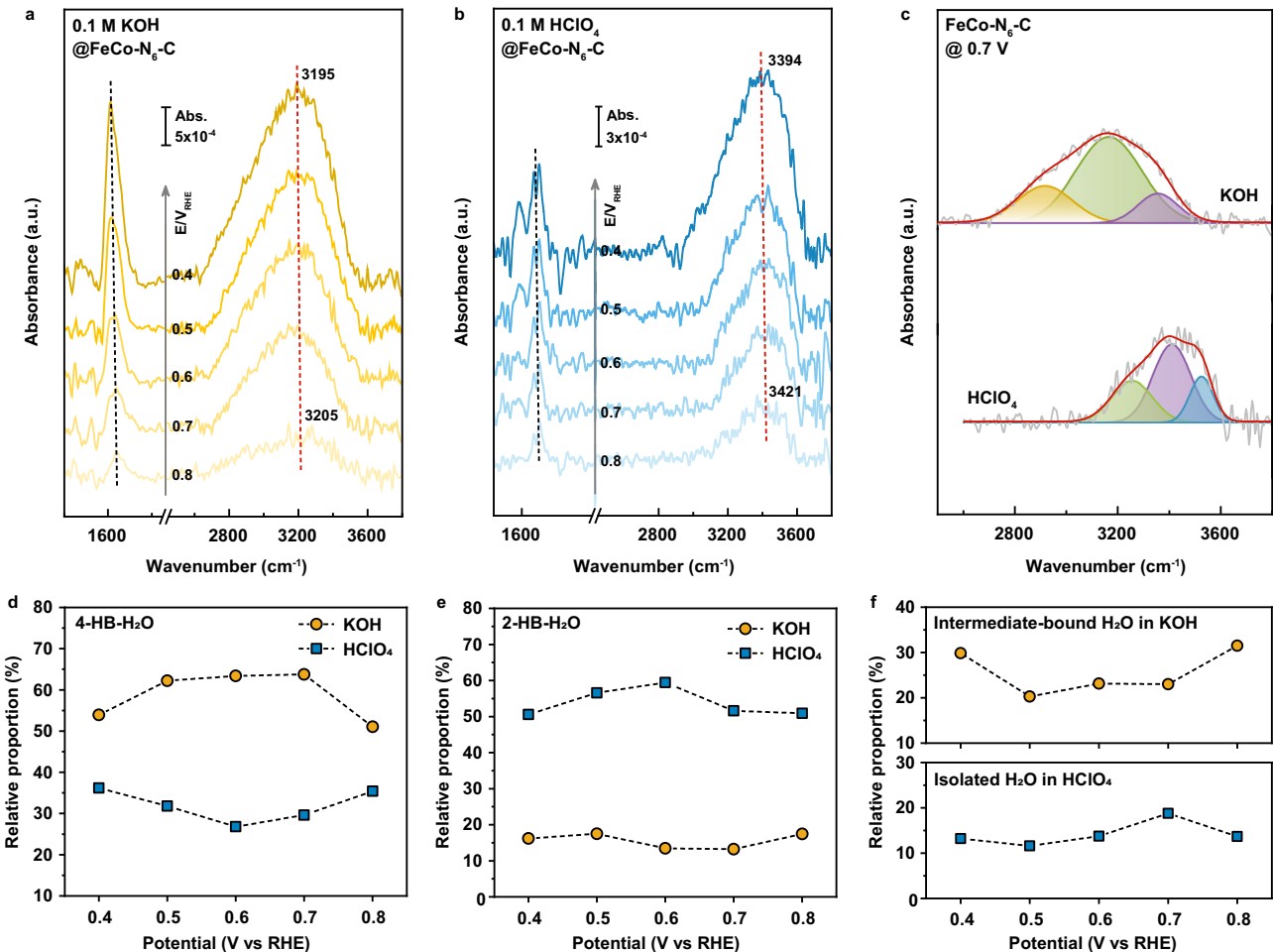

**Fig. 3 | SEIRAS characterization of double-layer microenvironments.** In situ SEIRAS spectra of ORR on FeCo-N$_6$-C recorded at potentials from 0.8 V to 0.4 V vs RHE in O$_2$-saturated solutions of (**a**) 0.1 M KOH and (**b**) 0.1 M HClO$_4$, where the reference spectrum was taken at 1.1 V. **c** Deconvolution of the O-H stretching peak at 0.7 V into three components. **d**–**f** Relative proportions of various kinds of interfacial water.

schematic diagrams of the determined ORR pathways at alkaline and acid interfaces, respectively, and the corresponding structures and reaction equations are illustrated in Fig. 4c, Supplementary Figs. 10 and 11. At alkaline interface, the O$_2$ molecule first adsorbs on the Fe-Co bridge site by replacing the adsorbed water molecules on Fe top site and meanwhile spontaneously dissociates into two *O intermediates (I → II in Fig. 4a); whereafter, the two *O intermediates on the Co and Fe top sites alternately obtain protons from interfacial water molecules to transform into two *OH intermediates that remain the top adsorption, accompanied by the electron transfer (PCET, II → III → IV); finally, the two *OH intermediates on the Co and Fe top sites continue to alternately obtain protons from interfacial water molecules to generate two water molecule products and back to initial state (PCET, IV → V → I). Such alkaline ORR pathway is in line with that identified through in situ synchrotron radiation Fourier transform infrared spectroscopy (SR-FTIR) recently reported by Liu et al.[32], but it should be recognized that the spontaneous dissociation of O$_2$ molecule benefits from the assistance of hydrogen bonds from interfacial water, rather than merely the presence of adjacent bimetallic sites. By contrast, although the first step at acid interface is also the replacement of adsorbed water molecule by O$_2$ molecule (I → II in Fig. 4b), the O$_2$ molecule does not undergo dissociation, and the subsequent reaction pathway is significantly different from that in alkaline. Specifically, at acid interface, the *O atom on Fe site preferentially obtains two protons from the interfacial water via PCET to generate a water molecule product (II → III → IV); then, the remaining *O atom on Fe-Co bridge site undergoes

the same PCET processes and back to the initial state (IV → V → I). In addition, it is noted that the O-O bond of adsorbed O$_2$ molecule also spontaneously breaks once it obtains a proton (II → III in Fig. 4b), rather than generating the *OOH intermediate.

What's more, it can be found that at each reaction intermediate state along the ORR pathway, the difference of double-layer microenvironments at alkaline and acid interfaces always maintains well. Namely, the interfacial water molecules around reactive center are orientated disorderly at alkaline interface while orderly in the form of O-down configuration at acid interface (Fig. 4c and Supplementary Fig. 11a); correspondingly, there exist abundant hydrogen bonds between the O atoms of surface oxygenated intermediates and the H atoms of interfacial water only at various reaction intermediate states of alkaline ORR, as illustrated by the radial distribution function analyses (Supplementary Fig. 12). On the other hand, based on the AIMD simulations of the whole ORR processes, the free energy diagrams are constructed to understand the reaction thermodynamics at alkaline and acid interfaces (see Supplementary Note 3 for details). As shown in Fig. 4d, the last PCET step (V → I in Fig. 4a) is the PDS of alkaline ORR; while for acid ORR, the PDS is the third PCET step (IV → V in Fig. 4b). Most notably, the free energy change of PDS at acid interface is much smaller than that at alkaline interface (1.16 eV vs 1.43 eV), which seems to suggest that the ORR activity should be more outstanding in acid than in alkaline. Nevertheless, the experimental phenomenon is against this suggestion, demonstrating that the dramatic activity gap between alkaline and acid ORR cannot be deciphered from the

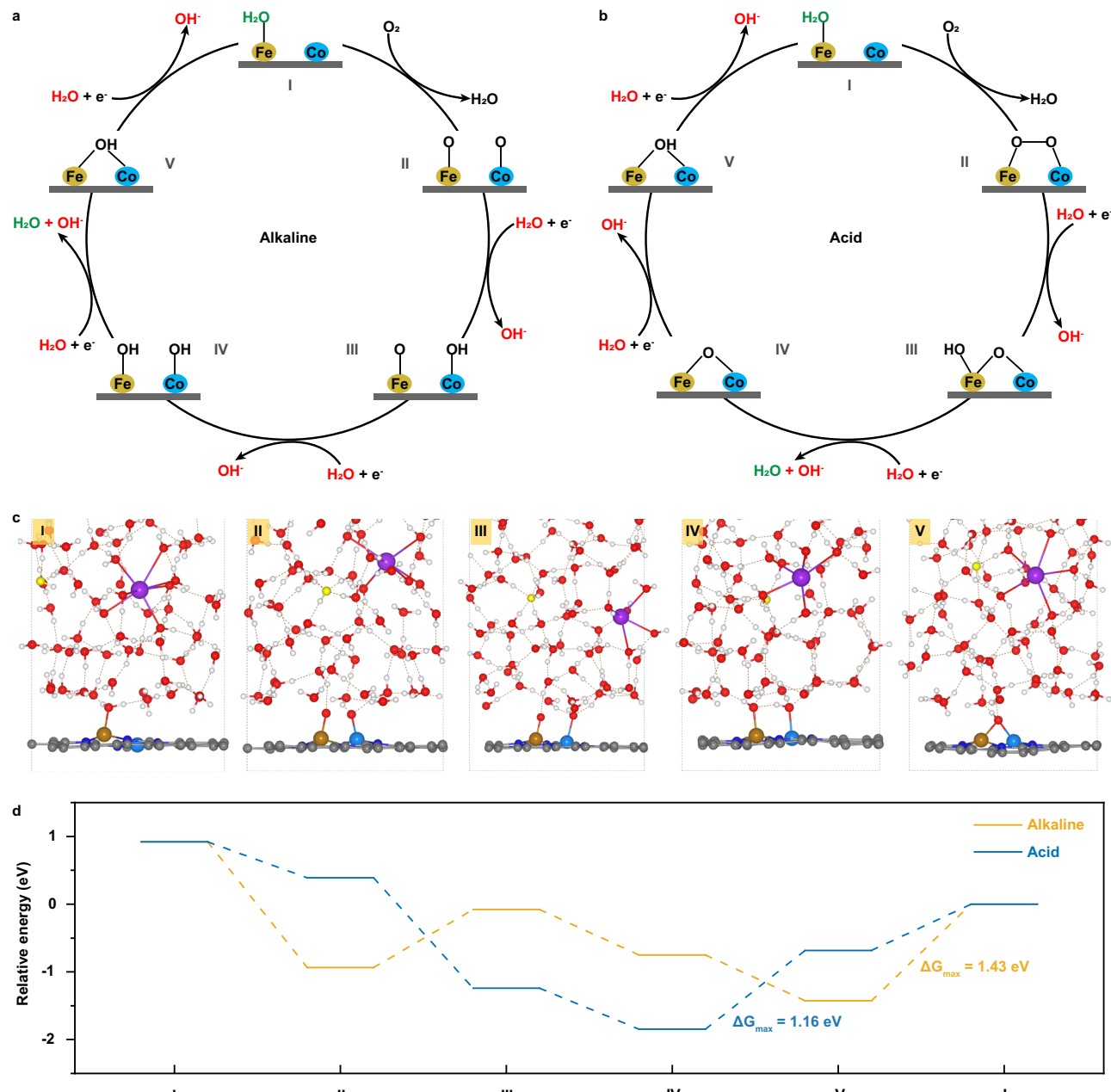

**Fig. 4 | ORR pathways and reaction energetics.** Schematic diagrams of the determined ORR mechanisms at alkaline (**a**) and acid (**b**) interfaces. The H$_2$O marked by red color represent the water molecules which serve as the proton donors and provide H atoms to surface oxygenated intermediates, while the H$_2$O marked by green color mean the reaction products. **c** Representative snapshots of the interfacial structures along the ORR process in alkaline media. **d** Free energy diagrams for ORR at alkaline and acid interfaces for $U = 1.0$ V vs RHE.

perspective of the energetics of multistep reaction pathways, but likely is a kinetic effect, which agrees with the speculation by Li et al. [33].

To further examine the accuracy and universality of the above results, we have also considered the existence of oxygenated spectators on FeCo-N$_6$-C catalyst, including the adjacent oxygenated spectators (shortened to Ad-OS)[34,35] and the axial oxygenated spectators (shortened to Ax-OS)[14,15,36–38]. The exact Ad-OS at alkaline and acid interfaces are identified as *OH$_{Ad-OS}$ and *O$_{Ad-OS}$ species occupying the Fe-Co bridge sites, respectively (Supplementary Fig. 13). Figure 5a, b show the FeCo-N$_6$-C/electrolyte interfaces with Ad-OS under alkaline and acid ORR conditions, in which the reactant O$_2$ molecules are adsorbed on the Fe top sites in the end-on configuration, due to the occupation of Fe-Co bridge sites by the Ad-OS and the stronger affinity of Fe element for oxygenated species. Figure 5c, d show the FeCo-N$_6$-

C/electrolyte interfaces with Ax-OS and *O$_2$, in which the Ax-OS species under alkaline and acid ORR conditions are also considered as *OH and *O, respectively. It is worth mentioning that despite the presence of Ad-OS or Ax-OS, one K$^+$ cation and one OH$^-$ anion also need to be introduced to reach the alkaline ORR potentials (-1.00 V and -1.05 V, respectively), and one H$_3$O$^+$ cation and two F$^-$ anions are required to attain the acid ORR potentials (-0.80 V and -0.90 V, respectively), which means that the electrode surfaces are still almost uncharged in alkaline while positively charged in acid. Correspondingly, the alkaline and acid interfaces still exhibit the same difference in the double-layer microenvironment features (Supplementary Figs. 14 and 15), compared to that without the oxygenated spectators (Fig. 2). Supplementary Fig. 16 and 17 show the determined ORR mechanisms in the presence of oxygenated spectators and the corresponding interface

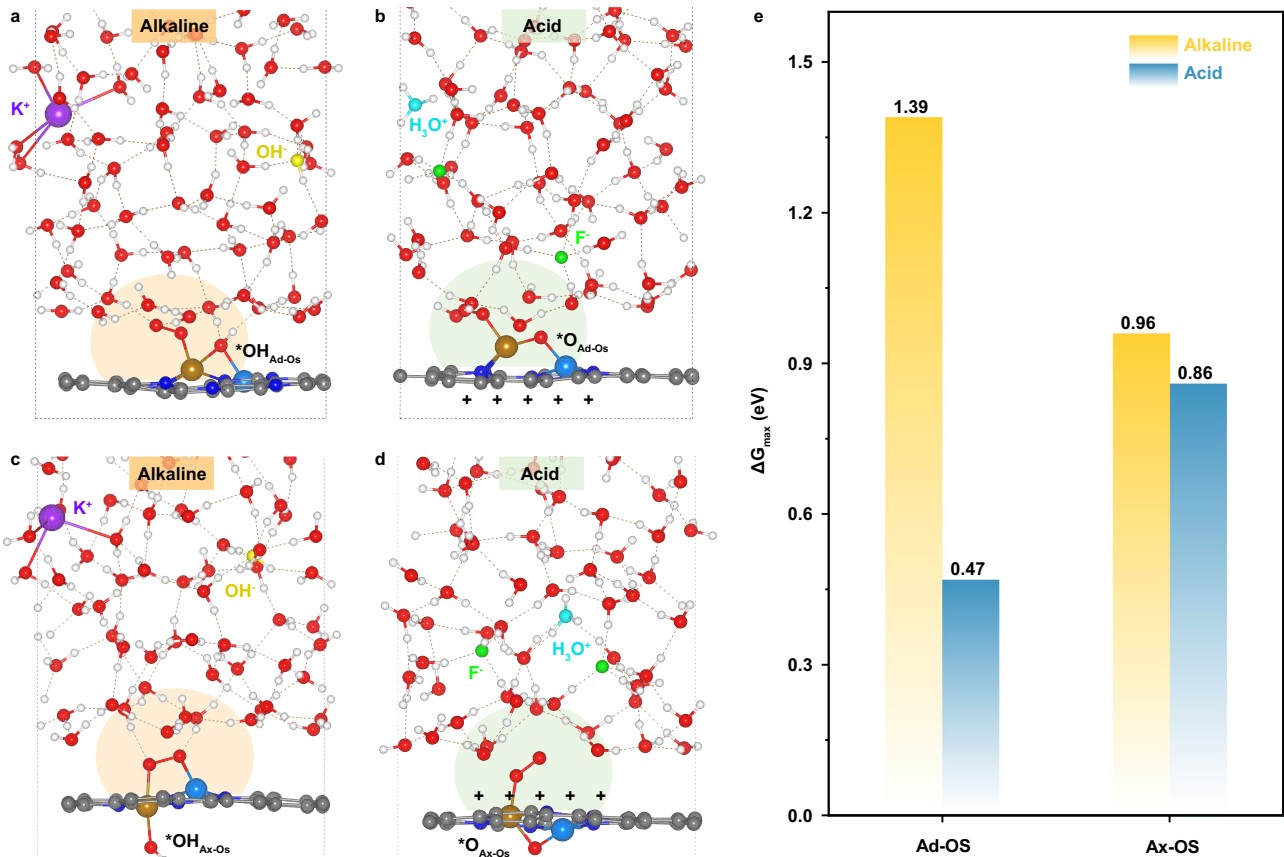

**Fig. 5 | Universality in the presence of oxygenated spectators.** Representative snapshots of the O₂ adsorbed interfaces at ORR potentials in alkaline and acid electrolytes with the existence of (**a**, **b**) adjacent oxygenated spectators (shortened to Ad-OS) and (**c**, **d**) axial oxygenated spectators (shortened to Ax-OS) on FeCo-N₆-

C. **e** Comparison of the maximum free energy change ($\Delta G_{max}$) in the ORR free energy diagrams at alkaline and acid interfaces when Ad-OS and Ax-OS are considered.

structures of various reaction intermediate states. It is noted that at both alkaline and acid interfaces with the Ad-OS, ORR proceed via the associative mechanism involving *OOH species, due to the destruction of synergistic effect between adjacent metal active sites by the spectators and the resulting end-on adsorption of O₂ molecule on Fe top site; while in the presence of Ax-OS, the alkaline and acid ORR proceed via the dissociative and associative mechanisms, respectively. Moreover, at each ORR reaction intermediate state, the hydrogen bonds between the O atoms of surface reaction intermediates and the H atoms of interfacial water can be observed clearly at alkaline interface, which is unlikely at acid interface (Supplementary Figs. 16c, d and 17c, d). Remarkably, the ORR free energy diagrams (Supplementary Figs. 16e and 17e) show that although the Ad-OS and Ax-OS species can affect the reaction energetics substantially, the PDS at alkaline interface is still much more uphill thermodynamically than that at acid interface (Fig. 5e), which confirms that the huge ORR activity gap between alkaline and acid media is not attributed to the difference in reaction energetics. All the above results laterally hint that the unfavorable interface double-layer microenvironment and the resultant sluggish PCET kinetics are very likely to be responsible for the inferior ORR performance in acid electrolyte.

## Origin of the dramatic ORR activity gap in alkaline and acid medias

To substantiate the above speculation, the kinetic free energy barriers for the first PCET reactions (II → III in Fig. 4a, b) in alkaline and acid ORR have been evaluated using the slow-growth sampling approach. The collective variable (CV) is defined as the distance between the O atom of surface oxygenated intermediate and the H atom of interfacial water

that is closest to the reactive O atom (Supplementary Fig. 18). At alkaline interface, two reaction processes for different water molecules that form hydrogen bonds with oxygenated intermediate are simulated; while at acid interface, three water molecules closest to the oxygenated intermediate are considered as the reactants. In addition, three independent slow-growth simulations for each reaction are performed to make the error analysis. Supplementary Fig. 19 shows the typical potential of mean force profiles derived from the slow-growth simulations for the first PCET reactions at alkaline and acid interfaces and the corresponding free energy profiles along the CV. Figure 6a–c indicate that the first PCET step in alkaline ORR involves two processes: the interfacial water molecule firstly provides its H atom to *O atom through the hydrogen bond, and meantime the generated OH⁻ anion turns into water molecule by receiving a H atom from another water molecule that locates further away from the electrode. By contrast, in addition to the above two processes, the first PCET step in acid ORR requires an additional process initially, namely the flip of water dipole orientation from O-down to H-down by overcoming the strong interfacial electric field (Supplementary Fig. 20). Resultantly, Fig. 6d and Supplementary Table 1 demonstrate that the kinetic free energy barriers for the first PCET reaction in acid ORR are much higher than that in alkaline ORR, thus contributing to the inferior ORR activity in acid electrolyte. Although the more actual barriers can be obtained when using the interface model with a SD closer to the experimental value, it has been stated that the SD would not change the differences of surface charge densities and interfacial microenvironments between alkaline and acid systems, as well as the relative sizes of the barriers of PCET steps. Therefore, these kinetic results not only unequivocally corroborate that the significant discrepancy of the interfacial double-layer microenvironments is the

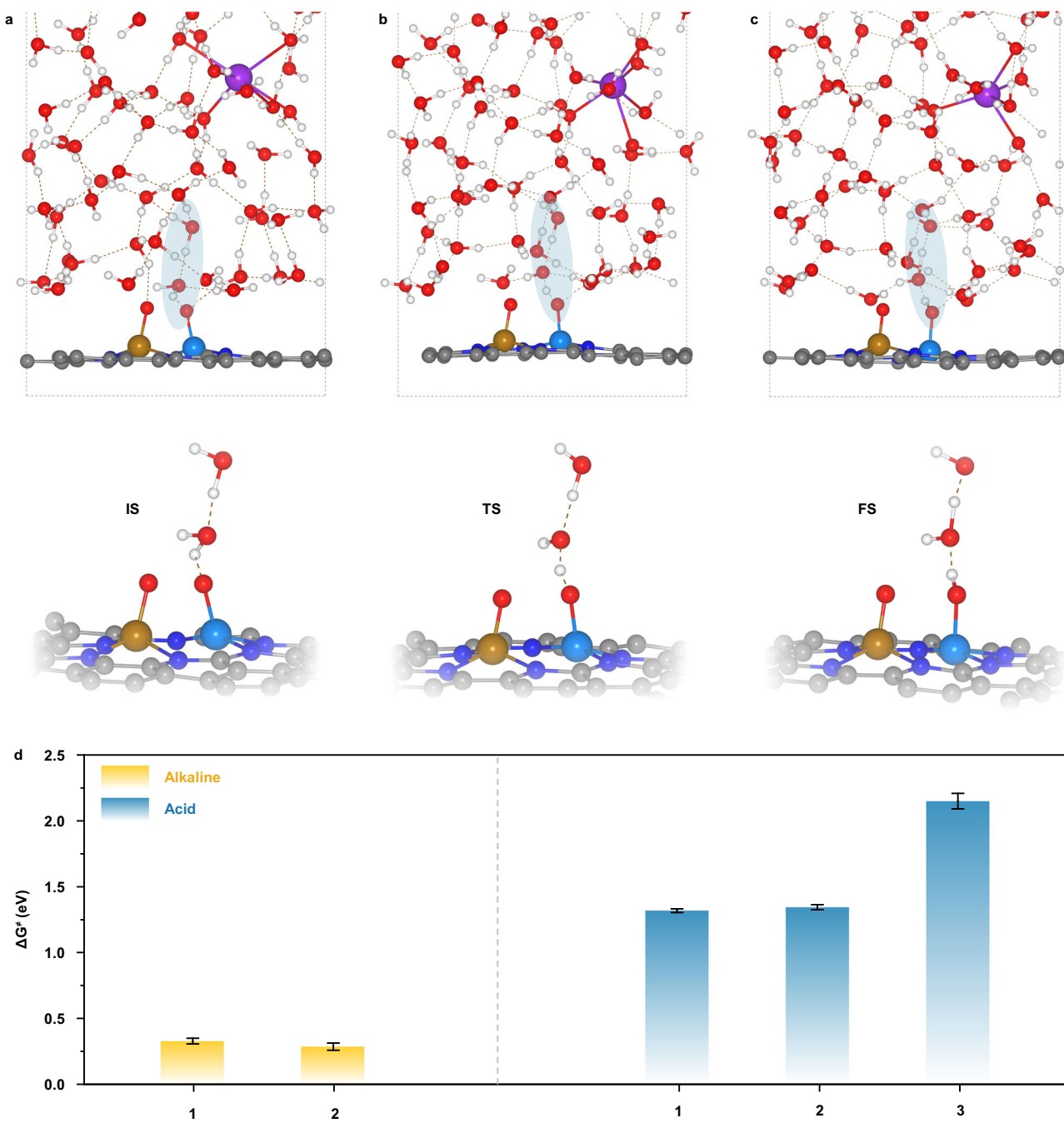

**Fig. 6 | Kinetic barrier calculations for the first PCET steps. a–c** Interface structures (upper panel) and corresponding close-up (lower panel) of the initial state (IS), transition state (TS) and final state (FS) in one typical slow-growth simulation for the first PCET reaction of alkaline ORR. The shadows represent the reaction domains. **d** Comparison of kinetic free energy barriers ($\Delta G^{\neq}$) for the first PCET reactions in alkaline and acid ORR. Error bars are obtained from the standard deviation of three independent simulations.

root of the dramatic ORR activity gap between alkaline and acid medias, but also highlight immense opportunities to improve the ORR kinetics in acid environment through the structural and functional modulations of electrocatalytic interface. In this regard, the introduction of protic ionic liquids and organic molecules and/or dispersing the charge accumulation on metal center through electrode structure design may be effective practice strategies to optimize the interfacial water orientation and construct the hydrogen bonds with surface oxygenated intermediates[39–41]. In addition, the scarcity of hydronium and the fairly high PCET barrier at acid interface caused by the much positively

charged electrode surface and ordered O-down water orientation imply that the protonation of FeCo-$N_6$ active center may be unlikely. Combined with the nonspecific adsorption property of perchlorate in $HClO_4$ electrolyte which is the system targeted in this study, the poisoning effects of anion and proton seems to play a rather minor role in inhibiting the ORR activity, which further hints the key role of disparate double-layer microenvironments in the pH effect of ORR on M-N-C catalysts. However, further studies are also required to holistically explore the possible effects of anions in acid electrolytes on ORR activity[42,43].

In summary, through combining the AIMD simulation, slow-growth sampling approach and in situ SEIRAS technology, the long-awaited origin of the dramatic ORR activity gap on M-N-C electrocatalysts between alkaline and acid medias has been explicitly and innovatively unveiled by taking the representative FeCo-$N_6$-C DAC as a model system. It is found that at the corresponding alkaline and acid ORR potentials, the surface charge states of FeCo-$N_6$-C electrode are quite different, thereby leading to the great distinction in the interfacial double-layer microenvironment, eventually rendering the pH-dependent ORR performance. Specifically, the FeCo-$N_6$-C electrode surface in alkaline media is almost uncharged, which results in the disorder arrangement of interfacial water around metal reactive moiety and thereby the natural formation of abundant hydrogen bonds between the O atoms of surface oxygenated intermediates and the H atoms of interfacial water molecules; while in acid media, the FeCo-$N_6$-C electrode is much more positively charged so that the interfacial water are orientated orderly in the O-down configuration and cannot form the hydrogen bonds with reaction intermediates. Due to the indispensable role of interfacial water as proton donors to provide hydrogen atoms to the ORR reaction intermediates in both alkaline and acid medias, the orderly orientated water molecules in O-down configuration and the resultant deficiency of hydrogen bonds at acid interface severely raise the kinetic barriers of PCET steps in ORR and then brings about the sluggish electrocatalytic activity. In addition, it is worth highlighting that the dramatic activity gap between alkaline and acid ORR cannot be explained from the perspective of energetics of multistep reaction pathways. Our study not only displays the fundamental importance of interfacial double-layer microenvironment in dominating the electrocatalytic reaction mechanism and kinetics, but also opens unique avenues for the design of high-efficiency M-N-C catalysts from the perspective of full optimization of the reaction environment.

## Methods

### Computations and models

All AIMD simulations were performed with the periodic density functional theory (DFT) program Vienna Ab initio Simulation Package[44]. The exchange-correlation energy was described by Perdew-Burke-Ernzerhof functional within general gradient approximation[45], and the electronic cores were treated by the projector augmented wave method[46]. The wave functions were expanded with a cutoff energy of 400 eV and the Gaussian smearing with a width of 0.1 eV was employed. To correct the van der Waals interaction, the zero damping DFT-D3 dispersion correction scheme of Grimme was applied[47]. No spin polarization is considered to reduce the computational cost because it has almost no impact on the overall energies[48–50]. A time step of 1 fs and only the gamma point of Brillouin zone with no consideration of symmetry are used for all AIMD simulations, and in which, the canonical ensemble condition (NVT) is imposed by a Nose-Hoover thermostat with a target temperature of 298 K during all simulations.

The FeCo-$N_6$-C/water interface is modeled by adding 67 water molecules above a FeCo-$N_6$-C catalyst slab with a SD of $-9.2 \times 10^{20}$ site/g, which is constructed from orthogonal ($3\sqrt{3} \times 4$) unit cells of graphene (Supplementary Fig. 1). The surface area is $-1.27$ nm$^2$ and the simulation box is $-4.2$ nm along the z axis, in which the thickness of the water film and the vacuum layer are $-1.6$ nm and $-2.4$ nm, respectively. It is noted that the SD in such model is much higher than that of the experimentally synthesized FeCo-$N_6$-C sample ($-1.1 \times 10^{20}$ site/g)[20], because the computational cost of AIMD simulation will be daunting if the SD in model is set to similar to experimental value. In this regard, the machine-learning potential based molecular dynamics simulations would play an important role in modeling the M-N-C material in real electrocatalytic conditions, although it is still fairly challenging currently[51,52]. For each AIMD simulation, an initial 5–10 ps of MD trajectory is used to adequately equilibrate the system, then followed by a production period of 10 ps. Snapshots of one in every 0.5 ps over the

final 10 ps AIMD trajectory are collected to calculate the mathematically averaged work function ($\Phi$) of the system, while the species concentration distribution profiles, water orientation profiles, radical distribution functions and bond length of adsorbed $O_2$ are statistically analyzed from the whole 10 ps AIMD production process. Inside, the work function ($\Phi$) of a system is calculated as the difference between vacuum level ($E_{vac}$) and fermi level ($E_F$), and $E_{vac}$ is extracted from the planar-averaged electrostatic potential profile, with $10^{-5}$ e/Å$^3$ being chosen as the cutoff value of the electron density[16,53]. Then, electrode potential versus reversible hydrogen electrode ($U_{RHE}$) can be calculated by the formula $U_{RHE} = (\Phi - \Phi_{SHE})/e + 0.059 \times pH$, where $\Phi_{SHE}$ is the work function of the standard hydrogen electrode and the value of 4.60 V is used for M-N-C systems in this work[15,54,55]; e is the electron charge; pHs are set to 1 for acid system and 13 for alkaline system, according to the used experimental electrolytes (0.1 M $HClO_4$ and 0.1 M KOH)[23]. The VDOS of interfacial water molecules was calculated by the Fourier transformation of the velocity auto-correlation functions in the AIMD trajectories.

To simulate the alkaline and acid electrolytes, we have introduced the potassium/hydroxide ions ($K^+$/$OH^-$) and hydronium/fluoride ($H_3O^+$/$F^-$) ions into the water film, respectively (Supplementary Fig. 2). Meanwhile, varying the net charge of the anion and cation pairs to regulate the surface charge density and thus the electrode potential to the experimental ORR potential, because the same amount of opposite charge as the net charge of ions will be applied to the FeCo-$N_6$-C electrode to keep the model charge neutral.

To simulate the first PCET steps in alkaline and acid ORR processes and calculate their activation free energy barriers, the slow-growth sampling approach within the constrained molecular dynamics framework is exploited[56]. The CV is defined as the distance between the H atom of reactive water and the O atom of adsorbed $O_2$, and the increment of 0.008 Å/fs for CV is set to derive the PCET reaction. For each reaction, three independent slow-growth simulations are performed to obtain the average and error bar of barrier. In addition, due to all simulations are performed under constant charge condition, the charge extrapolation method developed by Chan and Nørskov is used for constant potential corrections[57,58]. The detailed principles of the slow-growth method and constant potential correction can see the Supplementary Notes 1 and 2.

### Synthesis of FeCo-$N_6$-C catalyst

Firstly, the Zn/Co bimetallic MOF ($Zn_1Co_1$-BMOF) was synthesized. Specifically, 0.546 g $Co(NO_3)_2$·$6H_2O$ and 0.558 g $Zn(NO_3)_2$·$6H_2O$ were dissolved in 15 mL methanol, and then 0.616 g 2-methylimidazole (MeIM) in 15 mL methanol was subsequently injected into the above solution under vigorously stirring for 6 h at room temperature. The as-obtained precipitates were separated by centrifugation and washed with methanol several times and dried under vacuum at 70 °C for overnight. Then, to synthesize the FeCo-$N_6$-C catalyst, 100 mg $Zn_1Co_1$-BMOF was dispersed in 13 mL hexane by sonication for 1 h at room temperature; then 50 µL $FeCl_3$·$6H_2O$ (50 mg mL$^{-1}$) was added dropwise to the above solution under ultrasound for 10 min. The vial containing the slurry was stirred at room temperature for 2 h. The impregnated $Zn_1Co_1$-BMOF sample was separated by centrifugation and dried under vacuum at 70 °C for overnight. Finally, the powder of impregnated $Zn_1Co_1$-BMOF sample was placed in a tube furnace and then heated to the desired temperature (900 °C) for 2 h at a heating rate of 5 °C min$^{-1}$ under flowing Ar gas and then naturally cooled to room temperature to obtain the representative sample[20].

### Au thin film and working electrode preparation

The Au-coated Si hemispherical prism (25 mm in diameter) was used as the conductive substrate for catalysts and the IR reflection element. Detailed procedures on the Au thin film preparation was as follows. Before the chemical deposition of Au, the Si prism was first polished

gradually using 1.0, 0.3 and 0.05 μm $Al_2O_3$ powder until the surface became totally hydrophobic, and then sonicated and rinsed several times with acetone/ethanol mixed solution (volume ratio, v/v = 1:1) and double deionized-distilled water. Thereafter, the hydrophobic and clean Si prism underwent hydroxylation treatment with piranha solution (the volume ratio between concentrated $H_2SO_4$ and 30% $H_2O_2$ is 1:1) for 20 min and then was sonicated with distilled water several times. The infrared reflection plane of the Si prism was dried by Ar flow and treated by 40% $NH_4F$ for 2 min to make the surface be terminated with hydrogen. The Au thin film was then deposited on the hydrogen-terminated surface by immediately immersing this plane into a mixture of 9.2 mL Au plating solution and 124 μL 50% HF solution at 58 °C for 6 min[16].

The working electrode was prepared by dropping the catalyst suspension on the Au/Si surface with a loading of 0.19 mg cm$^{-2}$. The catalyst suspension (5 mg mL$^{-1}$) was prepared by dispersing the FeCo-$N_6$-C sample in 1 mL of mixed solution containing isopropyl alcohol and 5 vol % Nafion, and then ultrasonicated for 0.5 h to form a homogeneous ink.

### In situ ATR-SEIRAS experiments

The electrochemical ATR-SEIRAS measurements were carried out on a BRUKER INVENIO-R FTIR spectrometer equipped with a mercury cadmium telluride (MCT) detector cooled with liquid nitrogen in $O_2$-saturated aqueous solutions of 0.1 M KOH and 0.1 M $HClO_4$. Three-electrode configuration was utilized for electrochemical control, in which the Hg/HgO electrode and saturated calomel electrode (SCE) were used as the reference electrodes in alkaline and acid electrolytes, respectively; a Pt foil was used as the counter electrode. Unpolarized infrared radiation from an Elema source was focused at the interface at the incident angle of 60°, and the reflected radiation was detected. Real-time spectra with a resolution of 8 cm$^{-1}$ were recorded during stepping the working electrode potential, and each spectrum was integrated by 44 scans. All spectra are shown in the absorbance units $-\log(R/R_0)$, where R and $R_0$ represent the reflected intensity of the sample and reference spectrum, respectively. Reference spectra were taken at 1.1 V vs RHE.

## Data availability

All data supporting the findings of this study are available within the paper and its supplementary information files or from the corresponding authors upon request. Source data are provided with this paper.

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

## Acknowledgements

S.C. acknowledges the financial support from the National Natural Science Foundation of China (Nos. 21832004 and 22272122). P.L. acknowledges financial support from the National Natural Science Foundation of China (No. 22202156) and the China Postdoctoral Science Foundation (No. 2022M722452). Generous grants of computer time at the Wuhan Supercomputing Center and Supercomputing Center of Wuhan University are also gratefully acknowledged.

## Author contributions

S.C. supervised the project. P.L. and S.C. conceived the idea and designed the experiments. P.L. and Y.J. performed the AIMD simulations. P.L., H.F. and Y.M. performed the in situ SEIRAS experiments under the supervision of S.C. and C.G.; Y.R. performed the preparation of FeCo-N$_6$-C electrocatalyst under the supervision of Y.W.; P.L. and Y.J. analyzed the data. P.L., Y.J. and S.C. wrote the manuscript. All authors discussed the results and commented on the manuscript.

## Competing interests

The authors declare no competing interests.
