## [Peer Review File · Nature Communications]

Reviewers' comments:

Reviewer #1 (Remarks to the Author):

In the manuscript "Decisive Roles of Interfacial Water Orientation and Hydrogen Bonds in the Dramatic Activity Gap of Metal-Nitrogen-Carbon Catalysts for ORR in Alkaline and Acid", the authors carried out a computational investigation on the atomic details of electrified interfaces during oxygen reduction reaction (ORR) at non-platinum metal-nitrogen-carbon (MNC) catalyst. The authors particularly paid attention on understanding the origin of the distinct activity difference between alkaline and acidic ORR. The main claim of the authors is that the water structure formed at the acidic interface is more ordered with O-down configurations compared to the one formed at the alkaline interface. This leads the proton-coupled transfer to become more difficult at the acidic interface, increasing the barrier for the proton-coupled electron transfer (PCET) step. Also, the authors found that the bimetallic center of Fe and Co can spontaneously dissociate the O-O bond of dioxygen in the alkaline environment, which is contrary to the acidic case.

1. I think, the key and fundamental difference that the authors found between the two pH conditions are the different surface charge density at the same potential on a reversible hydrogen electrode (RHE) scale. Although the surface charge density is mostly determined by the potential difference from the point of zero charge (PZC), which is determined at the scale of standard hydrogen electrode (SHE), the potential at the RHE scale also includes a pH-dependent correction term of "0.059 pH". This leads to the alkaline interface being more negatively charged than the acidic interface at the same RHE scale, which not only facilitates an electron transfer from the electron-rich catalyst to the π^* bond of O_2 , resulting in an easy O-O bond break, but also attracts partial positive charges of water on H atoms to the catalyst surface, reducing the barrier for the proton transfer. This is a quite reasonable and interesting finding. However, the overall tone of the manuscript sounds a bit biased toward the different water orientations, as also appeared in the title. I think "different surface charge density" plays a more decisive role and "different water direction" is a result of surface charge density difference.

2. Also, the authors calculated the potential for two different interfaces when an O_2 molecule is adsorbed. Unless the lifetime of $*O_2$ is substantially long, and thereby the bimetallic sites are expected to be almost always covered with the $*O_2$, such a condition cannot be well rationalized. Probably, the potential should be defined using a surface where no adsorbate exists.

3. In Figure 3d, the authors calculated a reaction-free energy diagram. Are these calculated under a constant charge condition, or under a constant potential condition? The authors need to clarify this.

Reviewer #2 (Remarks to the Author):

In this interesting work from Chen and co-workers the ORR in alkaline and acidic media at FeCoN6C electrocatalyst is studied. The authors claim that the surface charge state depends on the pH. The interfacial double layer would be different depending on the pH and therefore the ORR would be different. The work is original and the state of the art is presented. The work is of interest to the field, but the authors address their claims with superficiality. The selected model electrocatalyst is not a common electrocatalyst. Real experiments are missing. Indeed local pH, surface charge, and water orientation can be measured but the authors are presenting only simulations. Also, the binding energy of cations and anions is not calculated and therefore the poisoning effect is not evaluated. The manuscript needs to be proofread. Because of all those major issues, my recommendation is to reject the work.

Reviewer #3 (Remarks to the Author):

In this work, the authors investigated the pH-dependence of ORR activity on FeCo-N6-C DAC by combining the AIMD simulation and slow-growth sampling approach, and demonstrated that the fundamental importance of interfacial double-layer microenvironment is vital when evaluating the reaction mechanism and kinetics. This work provides a new perspective addressing the pH-dependence of ORR activity; however, more detailed and solid demonstration should be given. Here are some comments.

1. Potential of zero charge (PZC) is a fundamental concept when talking about the structure of electric double layer, and the deviations from the PZC in alkaline and acidic media should be evaluated based on the correct determination of PZC. The authors should calculate the PZC of FeCo-N5-C DAC based on AIMD simulations and compare it with the experimental value.

2. About the identity of the cations, why was K used to construct the alkaline system, and what about Na or Cs?

3. About the spectator species. As many papers (10.1021/jacs.9b07712, 10.1021/jacs.2c08743, 10.1021/acscatal.2c00771, 10.1039/d1ta07791k) reported that, the spectator species in axially coordinated manner could affect the reaction energetics substantially, would the axially coordinated species like *O or *OH change the main conclusion?

4. More calculation details should be given.

(1) Regarding the slow-growth calculation, the potential of mean force profile should be given. The value of the error bars in Fig. 4 should be given.

(2) Regarding the free energy diagram calculation, what is the time scale or the number of configurations considered to calculate the statistically averaged internal energies of each reaction intermediate? What if the automatic hydrogenation of reaction intermediates in alkaline media?

5. Some errors or typos.

(1) The schematic diagram in step 2 is wrong in supplementary Fig. 4.

(2) H₂O in R-1 was missing in supplementary Fig. 5(b) and supplementary Fig. 6(c).

(3) The elementary reaction equations in supplementary Fig. 6(c) are not consistent with the reaction configurations in Fig. 3(b), supplementary Fig. 6(a) and (b).

(4) The second IS should be changed to FS in S18 in SI.

(5) There is a grammatical problem in the title. It might be "...in Alkaline and Acidic Electrolytes".

Responses to Reviewers' Comments

We sincerely appreciate the reviewers for the expert and valued comments and constructive suggestions which greatly help us to improve the quality of this submission. In the following, we carefully respond the reviewers' comments point by point (texts in blue), and describe the corresponding revisions made on the manuscript and Supplementary Information (orange font). The figures in this Response Letter are named as Figure R**.

Reviewer #1 (Remarks to the Author):

In the manuscript "Decisive Roles of Interfacial Water Orientation and Hydrogen Bonds in the Dramatic Activity Gap of Metal-Nitrogen-Carbon Catalysts for ORR in Alkaline and Acid", the authors carried out a computational investigation on the atomic details of electrified interfaces during oxygen reduction reaction (ORR) at non-platinum metal-nitrogen-carbon (MNC) catalyst. The authors particularly paid attention on understanding the origin of the distinct activity difference between alkaline and acidic ORR. The main claim of the authors is that the water structure formed at the acidic interface is more ordered with O-down configurations compared to the one formed at the alkaline interface. This leads the proton-coupled transfer to become more difficult at the acidic interface, increasing the barrier for the proton-coupled electron transfer (PCET) step. Also, the authors found that the bimetallic center of Fe and Co can spontaneously dissociate the O-O bond of dioxygen in the alkaline environment, which is contrary to the acidic case.

1. I think, the key and fundamental difference that the authors found between the two pH conditions are the different surface charge density at the same potential on a reversible hydrogen electrode (RHE) scale. Although the surface charge density is mostly determined by the potential difference from the point of zero charge (PZC), which is determined at the scale of standard hydrogen electrode (SHE), the potential at the RHE scale also includes a pH-dependent correction term of "0.059 pH". This leads to the alkaline interface being more negatively charged than the acidic interface at the same RHE scale, which not only facilitates an electron transfer from the electron-rich catalyst to the π^* bond of O_2 , resulting in an easy O-

O bond break, but also attracts partial positive charges of water on H atoms to the catalyst surface, reducing the barrier for the proton transfer. This is a quite reasonable and interesting finding. However, the overall tone of the manuscript sounds a bit biased toward the different water orientations, as also appeared in the title. I think "different surface charge density" plays a more decisive role and "different water direction" is a result of surface charge density difference.

Response 1: Thank the reviewer very much for this expert comment.

Yes, we fully agree with the reviewer's opinion that, the difference of surface charge densities at the same potential on RHE scale plays a decisive role in the dramatic ORR activity gap between alkaline and acid medias. While, the different interfacial water orientations are merely the result of surface charge density difference, which has been highlighted many times in the text of manuscript.

Inspired by this comment, we have put more emphasis on the fundamental importance of surface charge density difference in Title, Abstract and Conclusions parts. The corresponding descriptions in the revised manuscript are attached below:

In the Title (Lines 1-3 Page 1): "Decisive Role of Disparate Double-Layer Microenvironments Induced by Surface Charge Density Difference in the pH effect of ORR on Metal-Nitrogen-Carbon Catalysts"

In the Abstract (Lines 25-26 Page 2): "We conclude that the great charge density difference on electrode surface brings about disparate orientations of interfacial water under alkaline and acid ORR conditions"

In the Conclusions (Lines 369-371 Page 21): "the surface charge states of FeCo-N₆-C electrode are quite different, thereby leading to the great distinction in the interfacial double-layer microenvironment, eventually rendering the pH-dependent ORR performance."

2. Also, the authors calculated the potential for two different interfaces when an O₂ molecule is adsorbed. Unless the lifetime of *O₂ is substantially long, and thereby the bimetallic sites are expected to be almost always covered with the *O₂, such a condition cannot be well rationalized. Probably, the potential should be defined using a surface where no adsorbate exists.

Response 2: We thank the reviewer for the careful reading and incisive comments.

Firstly, we agree the reviewer that the lifetime of *O_2 intermediate may be indeed short on the bimetallic site. However, it can be realized that there should always exist oxygen-containing intermediates (e.g., *OOH , *OH , *O) along the whole ORR process. More importantly, due to the electron-withdrawing effect, the interfaces with these oxygen-containing intermediates often possess distinctly higher potentials comparing to that without any oxygen-containing intermediates (Figure R1). It means that under the ORR condition, the interfacial microenvironment will change from the bare system to the oxygen-containing adsorbed systems when the electrode potential is controlled at a same value. In other words, if we define the electrode potential using the surface where no adsorbate exists, the potentials of the subsequent surfaces with adsorbates should also be readjusted again to the same scenes in the manuscript. In addition, it can be noted that the electrode potentials of the interfaces with oxygen-containing intermediates are fairly similar, comparing to the bare interface (Figure R1). Considering the above reasons, we have defined the electrode potential using the interface with *O_2 intermediate, which is the first reaction intermediate state although the lifetime may be short.

Figure R1. The electrode potentials (U vs RHE) at different reaction intermediate states for acid system.

We have added the above discussions as the note of Supplementary Figure 2 in the revised Supplementary Information, and the revisions are attached below:

In Page S3 of Supplementary Information: "In this work, the definition and adjustment of the electrode potentials corresponding to the experimental ORR conditions are based on the interfaces with *O_2 (Fig. 1 in the text), rather than the clean interfaces where no adsorbates

exist as shown in Supplementary Fig. 2. This is mainly for two reasons. First, it can be realized that there should always exist oxygen-containing intermediates (e.g., *OOH, *OH, *O) along the whole ORR process. Secondly, due to the electron-withdrawing effect, the interfaces with these oxygen-containing intermediates often possess distinctly higher potentials comparing to that without any oxygen-containing intermediates. For example, the potentials of interfaces shown in Fig. 1b and Supplementary Fig. 2b are 0.88 V and 0.58 V, respectively. It means that under the ORR condition, the interfacial microenvironment will change from the bare system to the oxygen-containing adsorbed systems when the electrode potential is controlled at a same value. In other words, if we define the electrode potential using the interface where no adsorbate exists, the potentials of the subsequent surfaces with adsorbates should also be readjusted again. Therefore, we define the electrode potential using the interface with *O₂ intermediate, which is the first reaction intermediate state along ORR process.”

3. In Figure 3d, the authors calculated a reaction-free energy diagram. Are these calculated under a constant charge condition, or under a constant potential condition? The authors need to clarify this.

Response 3: Thank the reviewer very much for reminding us this important point.

We are very sorry for the unclear description. All free energy diagrams for ORR are calculated under the constant charge condition. On this basis, two correction terms have been added for each elementary reaction. Related approaches have been provided in the Supplementary Notes 1 and 3. More detailed descriptions are as follows.

Due to the electrode potential change at different reaction intermediate states along the ORR process, the first one is the constant potential correction term $\pm \frac{(q_2 - q_1)(\Phi_2 - \Phi_1)}{2}$ proposed by Chan and Nørskov (*J. Phys. Chem. Lett.* 2015, 6, 2663-2668; *J. Phys. Chem. Lett.* 2016, 7, 1686-1690), in which $\Phi_2 - \Phi_1$ and $q_2 - q_1$ are the changes in work function and charge of the FeCo-N₆-C electrode plus all adsorbates, respectively, from state 1 to state 2, to make the calculation of reaction free energy diagram as close as possible to the constant potential condition. The '+' and '-' symbols represent that the corrections are performed at constant potential Φ_1 and Φ_2 , respectively, which is closer to the work function of *O₂ reaction intermediate state (viz. the state I shown in Figure 4a,b of the manuscript) is chosen as the

constant potential. It is worth mentioning that such a correction value (ΔE_{corr}) for each elementary step is fairly small and even can be neglected (Figure R2).

Figure R2. The constant potential correction values (ΔE_{corr}) along ORR pathways in alkaline and acid medias.

The second correction term is the $e(U - U_{\text{cat}})$, in which U is the target potential (1.0 V vs RHE) and U_{cat} is the actual potential of AIMD simulated acid or alkaline system (viz. 0.88 V or 1.10 V vs RHE), so as to compare the reaction free energy diagrams in alkaline and acid at the same electrode potential. Figure R3 shows the reaction free energy diagrams of alkaline and acid ORR before this correction, which displays the same conclusions with that shown in Figure 4d in the manuscript, namely the free energy change of potential determined step PDS at acid interface is much smaller than that at alkaline interface (1.04 eV vs 1.53 eV).

Figure R3. Free energy diagrams for ORR at alkaline interface for $U = 1.10$ V and acid interface for $U = 0.88$ V vs RHE.

We have added more details into the Supplementary Note 3 (Pages S29-S30) in the revised Supplementary Information.

Reviewer #2 (Remarks to the Author):

In this interesting work from Chen and co-workers the ORR in alkaline and acidic media at FeCoN₆C electrocatalyst is studied. The authors claim that the surface charge state depends on the pH. The interfacial double layer would be different depending on the pH and therefore the ORR would be different. The work is original and the state of the art is presented. The work is of interest to the field, but the authors address their claims with superficiality. The selected model electrocatalyst is not a common electrocatalyst. Real experiments are missing. Indeed local pH, surface charge, and water orientation can be measured but the authors are presenting only simulations. Also, the binding energy of cations and anions is not calculated and therefore the poisoning effect is not evaluated. The manuscript needs to be proofread. Because of all those major issues, my recommendation is to reject the work.

Response: Thank the reviewer very much for bring these important issues to our attention. The following is our response point by point.

(1) For the comment "The selected model electrocatalyst is not a common electrocatalyst."

In recent years, dual-atom catalysts (DACs) have attracted extensive attention, as an extension of single-atom catalysts (SACs). Compared with SACs, DACs have higher metal loading and more complex and flexible active sites, thus achieving better catalytic performance and providing more opportunities for energy-related electrocatalysis reactions (*Nat. Commun.* 2023, 14, 291; *Nat. Energy* 2021, 6, 1054-1066; *Adv. Mater.* 2021, 2102576; *Sci. Adv.* 2020, 6, eaba6586; *J. Am. Chem. Soc.* 2019, 141, 17763-17770; *Adv. Mater.* 2019, 31, 1905622; *J. Am. Chem. Soc.* 2018, 140, 10757-10763; *J. Am. Chem. Soc.* 2017, 139, 17281-17284; etc.). It can be believed that the DACs have a fairly broad application prospect in energy conversion and storage technologies, such as fuel cells, electrocatalytic CO₂ reduction and so on. Thereinto, FeCo-N-C DAC catalyst has often been demonstrated as the most highly efficient ORR electrocatalysts (*Adv. Mater.* 2021, 2102576; *ACS Catal.* 2020, 10, 2754-2761; *Adv. Funct. Mater.* 2020, 2007423; etc.). Therefore, it can be considered that the choice of FeCo-N₆-C in this manuscript is common and representative.

Furthermore, in light of this insightful comment, we have also expanded the scope of our studies to the single atom catalysts (SACs), including the Fe-N₄-C and Co-N₄-C. We have calculated the potentials of zero free charge (PZFCs) of the Fe-N₄-C/H₂O and Co-N₄-C/H₂O

interfaces shown in Figure R4a,b, and the potentials of zero total charge (PZTCs) of these interfaces with adsorbed oxygen molecules (*O_2) in end-on configuration shown in Figure R4c,d. The calculated PZFCs for Fe-N₄-C/H₂O and Co-N₄-C/H₂O systems are -0.71 V and -0.34 V vs SHE, respectively, both being similar to the values calculated by Karen Chan and Liu et al (*J. Am. Chem. Soc.* 2021, 143, 9423-9428; *ACS Catal.* 2020, 10, 7826-7835). And the calculated PZTCs for Fe-N₄-C/H₂O and Co-N₄-C/H₂O systems with *O_2 are 0.17 V and 0.37 V vs SHE, respectively. Notably, Figure R4e,f show that both the PZTCs of Fe-N₄-C/H₂O and Co-N₄-C/H₂O systems are significantly lower than the ORR reaction potential at pH=1 while fairly close to the ORR potential at pH=13. It means that the electrodes are much positively charged in acid and almost uncharged in alkaline, which is fully consistent with the FeCo-N₆-C/H₂O system. Therefore, it can be believed that the conclusions in the manuscript are universal for the SACs, namely the great difference of interfacial microenvironments in alkaline and acid induced by the different surface charge densities plays a key role in determining the dramatic activity gap of M-N-C catalysts for ORR.

Figure R4. (a,b) Representative snapshots of Fe-N₄-C/H₂O and Co-N₄-C/H₂O interfaces. (c,d) Representative snapshots of Fe-N₄-C/H₂O and Co-N₄-C/H₂O interfaces with *O_2 . The Fe, Co, N, C, O and H atoms are colored with brown, sky blue, blue, gray, red and white, respectively. The brown dashed lines represent the hydrogen bonds. (e,f) Pourbaix diagram showing the pH dependence of the ORR reaction potential (1.0 V vs RHE is used here), the PZFC and PZTC

for Fe-N₄-C/H₂O and Co-N₄-C/H₂O systems.

We have added Figure R4 as Supplementary Figure 4 in the revised Supplementary Information. Meanwhile, the corresponding descriptions have been added in the revised manuscript, and the revisions are attached below:

In Lines 168-173 Page 9: “In addition, it is worth mentioning that such scenarios are also likely to exist for SACs. As shown in Supplementary Fig. 4, it is clear that both potentials of zero total charge (PZTCs) of the Fe-N₄-C/water and Co-N₄-C/water interfaces with *O₂ are distinctly lower than the ORR potential at pH=1 while fairly close to the ORR potential at pH=13, implying that the SAC electrodes are also much positively charged in acid and almost uncharged in alkaline.”

(2) For the comment “Real experiments are missing. Indeed local pH, surface charge, and water orientation can be measured but the authors are presenting only simulations.”

Yes, we fully agree the reviewer’s opinion that the experimental validation is important for the simulated conclusions. Inspired by this insightful suggestion, we have collaborated with Prof. Yuen Wu in the University of Science and Technology of China, the developer of the FeCo-N₆-C catalyst, to obtain the catalyst sample, and then performed the in situ attenuated total reflectance surface-enhanced infrared absorption spectroscopy (ATR-SEIRAS) experiments to measure the interfacial water orientation information at alkaline and acid interfaces. The preparation detail and structure characterizations of the FeCo-N₆-C sample can be seen in our previous report (*J. Am. Chem. Soc.* 2017, 139, 17281-17284). Such spectroscopic results can also be believed to a certain extent to represent the differences in surface charge density and local pH, because these factors are generally interrelated with each other.

Figure R5a,b are the in situ SEIRAS spectra of ORR on FeCo-N₆-C catalyst recorded at potentials from 0.8 V to 0.4 V vs RHE in O₂-saturated solutions of 0.1 M KOH and 0.1 M HClO₄, which clearly show the O-H stretching mode (~2800-3600 cm⁻¹) and H-O-H bending mode (~1600-1700 cm⁻¹) of interfacial water. It is noted that both the frequencies and intensities of these two peaks exhibit an obvious potential dependence, suggesting that the obtained spectral signals are primarily derived from the first few water layers close to the electrode surface (*J. Am. Chem. Soc.* 2005, 127, 15916-15922). More importantly, the O-H stretching peaks at various potentials in alkaline electrolytes mainly locate around ~3200 cm⁻¹, which is redshifted

by 200 cm^{-1} compared to that in acid media ($\sim 3400\text{ cm}^{-1}$). This implies that the interfacial water molecules in alkaline environment are strongly hydrogen-bonded due to their disordered orientation on an almost uncharged electrode surface and similarity to the bulk water. By contrast, the interfacial water molecules in acid electrolyte are orderly arranged in O-down configurations on a much positively charged electrode surface, which brings about the decrease in the number of hydrogen bonds of interfacial water molecules and thus higher O-H stretching vibration frequency.

Moreover, the O-H stretching peaks are further deconvoluted into three distinct components through Gaussian fitting, as shown in Figure R5c for 0.7 V and Figure R6 for other potentials, corresponding to three types of O-H stretching vibrations. Thereinto, the components locating at $\sim 3400\text{ cm}^{-1}$ (green shadow) and $\sim 3200\text{ cm}^{-1}$ (purple shadow) exist in both alkaline and acid electrolytes and are assigned to the 4-coordinated hydrogen-bonded water (4-HB-H₂O) and 2-coordinated hydrogen-bonded water (2-HB-H₂O), respectively (*Nature* 2021, 600, 81-85; *Science* 2001, 292, 908-912; *Nat. Mater.* 2019, 18, 697-701; *ACS Nano* 2014, 8, 2704-2713; *JACS Au* 2021, 1, 1674-1687; *J. Phys. Chem. Lett.* 2010, 1, 1487-1491; *J. Phys. Chem. C* 2008, 112, 4248-4256; et al.). Figure R5d,e show that the relative proportion of 4-HB-H₂O in alkaline media ($\sim 59\%$) is much higher than that in acid electrolyte ($\sim 32\%$), while the relative proportion of 2-HB-H₂O is opposite ($\sim 16\%$ vs $\sim 54\%$), which imply that the interfacial water molecules are indeed with stronger hydrogen-bond interactions in alkaline system. Besides, it is found that there also exists another component possessing a fairly low frequency of $\sim 2950\text{ cm}^{-1}$ (yellow shadow in Figure R5c) and an average relative proportion of $\sim 25\%$ (upper panel in Figure R5f) in alkaline system, which may correspond to the interfacial water molecules that forms hydrogen-bonds with the oxygenated intermediates (termed as intermediate-bound H₂O). To support this conjecture, the computational vibrational density of states (VDOS) of intermediate-bound H₂O at the simulated alkaline interface is calculated. Figure R7 shows that the VDOS of intermediate-bound H₂O does contribute obvious vibration peaks located from 2800 cm^{-1} to 3000 cm^{-1} , and meanwhile the strongly intermediate-bound water molecules with five hydrogen bonds can be directly observed along the AIMD trajectory (inset of Figure R7), which confirms the accuracy of the above assignment for the low frequency component and provides a solid support for the alkaline double-layer microenvironment. Conversely, the third component in acid

electrolyte possesses a quite high frequency of $\sim 3550\text{ cm}^{-1}$ (blue shadow in Figure R5c) and an average relative proportion of only $\sim 14\%$ (lower panel in Figure R5f), which is associated with the isolated H_2O that may form due to the further destroy of hydrogen-bond interaction exerted by the ions in double-layer (*Nature* 2021, 600, 81-85; *JACS Au* 2021, 1, 1674-1687; et al.). To sum up, it can be seen that the salient consistency between the experimental spectroscopy, computational spectroscopy and AIMD simulation results could unequivocally substantiate the discrepancy of the double-layer microenvironments at alkaline and acid ORR interfaces.

Figure R5. (a,b) In situ SEIRAS spectra of ORR on FeCo-N₆-C recorded at potentials from 0.8 V to 0.4 V vs RHE in O₂-saturated solutions of (a) 0.1 M KOH and (b) 0.1 M HClO₄, where the reference spectrum was taken at 1.1 V. (c) Deconvolution of the O-H stretching peak at 0.7 V into three components. (d-f) Relative proportions of various kinds of interfacial water.

Figure R6. Deconvolution of O-H stretching features of in situ SEIRAS spectra (grey curves) of ORR on FeCo-N₆-C recorded from 0.8 V to 0.4 V vs RHE in O₂ saturated (a) 0.1 M KOH and (b) 0.1 M HClO₄ solutions. Spectra were subtracted by the reference spectrum taken at 1.1 V vs RHE. The OH stretching peaks were deconvoluted into three components in both alkaline and acid systems.

Figure R7. The computational VDOS (grey curve) of the OH stretching feature of interfacial water molecules that form hydrogen bonds with the surface oxygenated intermediates at O₂ adsorbed alkaline interface. The Gaussian fitting has been performed.

We have added Figure R5 as Fig. 3 in the revised manuscript, added Figure R6 and R7 as Supplementary Figure 5 and Supplementary Figure 6, respectively, in the revised Supplementary Information. Meanwhile, the corresponding descriptions have been added in the revised manuscript, and the revisions are attached below:

In Lines 20-21 Page 1: “by combining the ab initio molecular dynamics simulation, the slow-growth enhanced free-energy sampling approach and in situ surface-enhanced infrared absorption spectroscopy”

In Lines 74-75 Page 4: “and in situ surface-enhanced infrared absorption spectroscopy (SEIRAS) with the attenuated total reflection (ATR) configuration”

In Lines 176-227 Pages 9-12: **“In situ SEIRAS verification of the double-layer microenvironment difference.** To experimentally rationalize the AIMD-simulated double-layer microenvironments in alkaline and acid electrolytes and their difference, the in situ SEIRAS measurements are performed to probe the interfacial water structures. Figure 3a,b are the in situ SEIRAS spectra of ORR on FeCo-N6-C catalyst recorded at potentials from 0.8 V to 0.4 V vs RHE in O₂-saturated solutions of 0.1 M KOH and 0.1 M HClO₄, which clearly show the O-H stretching mode (~2800-3600 cm⁻¹) and H-O-H bending mode (~1600-1700 cm⁻¹) of interfacial water. It is noted that both the frequencies and intensities of these two peaks exhibit an obvious potential dependence, suggesting that the obtained spectral signals are primarily derived from the first few water layers close to the electrode surface²⁵. More importantly, the O-H stretching peaks at various potentials in alkaline electrolytes mainly locate around ~3200 cm⁻¹, which is redshifted by 200 cm⁻¹ compared to that in acid media (~3400 cm⁻¹). This implies that the interfacial water molecules in alkaline environment are strongly hydrogen-bonded due to their disordered orientation on an almost uncharged electrode surface and similarity to the bulk water. By contrast, the interfacial water molecules in acid electrolyte are orderly arranged in O-down configurations on a much positively charged electrode surface, which brings about the decrease in the number of hydrogen bonds of interfacial water molecules and thus higher O-H stretching vibration frequency.

Moreover, the O-H stretching peaks are further deconvoluted into three distinct components through Gaussian fitting, as shown in Fig. 3c for 0.7 V and Supplementary Fig. 5 for other potentials, corresponding to three types of O-H stretching vibrations. Thereinto, the components locating at ~3400 cm⁻¹ (green shadow) and ~3200 cm⁻¹ (purple shadow) exist in both alkaline and acid electrolytes and are assigned to the 4-coordinated hydrogen-bonded water (4-HB-H₂O) and 2-coordinated hydrogen-bonded water (2-HB-H₂O), respectively²⁶⁻³¹. Figure 3d,e show that the relative proportion of 4-HB-H₂O in alkaline media (~59%) is much

higher than that in acid electrolyte (~32%), while the relative proportion of 2-HB-H₂O is opposite (~16% vs ~54%), which imply that the interfacial water molecules are indeed with stronger hydrogen-bond interactions in alkaline system. Besides, it is found that there also exists another component possessing a fairly low frequency of ~2950 cm⁻¹ (yellow shadow in Fig. 3c) and an average relative proportion of ~25% (upper panel in Fig. 3f) in alkaline system, which may correspond to the interfacial water molecules that forms hydrogen-bonds with the oxygenated intermediates (termed as intermediate-bound H₂O). To support this conjecture, the computational vibrational density of states (VDOS) of intermediate-bound H₂O at the simulated alkaline interface is calculated. Supplementary Figure 6 shows that the VDOS of intermediate-bound H₂O does contribute obvious vibration peaks located from 2800 cm⁻¹ to 3000 cm⁻¹, and meanwhile the strongly intermediate-bound water molecules with five hydrogen bonds can be directly observed along the AIMD trajectory (inset of Supplementary Fig. 6), which confirms the accuracy of the above assignment for the low frequency component and provides a solid support for the alkaline double-layer microenvironment. Conversely, the third component in acid electrolyte possesses a quite high frequency of ~3550 cm⁻¹ (blue shadow in Fig. 3c) and an average relative proportion of only ~14% (lower panel in Fig. 3f), which is associated with the isolated H₂O that may form due to the further destroy of hydrogen-bond interaction exerted by the ions in double-layer^{26,29}. To sum up, it can be seen that the salient consistency between the experimental spectroscopy, computational spectroscopy and AIMD simulation results could unequivocally substantiate the discrepancy of the double-layer microenvironments at alkaline and acid ORR interfaces.”

In Lines 365-366 Page 20: “and in situ SEIRAS technology”

In Lines 420-422 Page 23: “The VDOS of interfacial water molecules was calculated by the Fourier transformation of the velocity auto-correlation functions in the AIMD trajectories.”

In Lines 439-458 Pages 24-25: “**In situ ATR-SEIRAS experiments.** The electrochemical ATR-SEIRAS measurements were carried out on a BRUKER INVENIO-R FTIR spectrometer equipped with a mercury cadmium telluride (MCT) detector cooled with liquid nitrogen in O₂-saturated aqueous solutions of 0.1 M KOH and 0.1 M HClO₄. Three-electrode configuration was utilized for electrochemical control, in which the Hg/HgO electrode and saturated calomel electrode (SCE) were used as the reference electrodes in alkaline and acid electrolytes,

respectively; a Pt foil was used as the counter electrode. The Au-coated Si hemispherical prism (25 mm in diameter) was used as the conductive substrate for catalysts and the IR reflection element. Detailed procedures on the Au thin film preparation and instrument setup can be found in our previous study¹⁶. The working electrode was prepared by dropping the catalyst suspension on the Au/Si surface with a loading of 0.19 mg cm⁻². The catalyst suspension (5 mg mL⁻¹) was prepared by dispersing the FeCo-N₆-C sample in 1 mL of mixed solution containing isopropyl alcohol and 5 vol % Nafion, and then ultrasonicated for 0.5 h to form a homogeneous ink. The preparation detail and structure characterizations of the FeCo-N₆-C sample can be seen in our previous report²⁰. Unpolarized infrared radiation from an Elema source was focused at the interface at the incident angle of 60°, and the reflected radiation was detected. Real-time spectra with a resolution of 8 cm⁻¹ were recorded during stepping the working electrode potential, and each spectrum was integrated by 44 scans. All spectra are shown in the absorbance units $-\log(R/R_0)$, where R and R₀ represent the reflected intensity of the sample and reference spectrum, respectively. Reference spectra were taken at 1.1 V vs RHE.”

References: “25. Schultz, Z. D., Shaw, S. K. & Gewirth, A. A. Potential dependent organization of water at the electrified metal-liquid interface. *J. Am. Chem. Soc.* **127**, 15916-15922 (2005).
26. Wang, Y. H. et al. In situ Raman spectroscopy reveals the structure and dissociation of interfacial water. *Nature* **600**, 81-85 (2021).
27. Scatena, L. F., Brown, M. G. & Richmond, G. L. Water at hydrophobic surfaces: weak hydrogen bonding and strong orientation effects. *Science* **292**, 908-912 (2001).
28. Chen, H. C. et al. Active and stable liquid water innovatively prepared using resonantly illuminated gold nanoparticles. *ACS Nano* **8**, 2704-2713 (2014).
29. Huang B. et al. Cation- and pH-dependent hydrogen evolution and oxidation reaction kinetics. *JACS Au* **1**, 1674-1687 (2021).
30. Yamakata A. & Osawa M. Destruction of the water layer on a hydrophobic surface induced by the forced approach of hydrophilic and hydrophobic cations. *J. Phys. Chem. Lett.* **1**, 1487-1491 (2010).
31. Osawa M. et al. Structure of water at the electrified platinum-water interface: A study by surface-enhanced infrared absorption spectroscopy. *J. Phys. Chem. C* **112**, 4248-4256 (2008).”

(3) For the comment “Also, the binding energy of cations and anions is not calculated and therefore the poisoning effect is not evaluated.”

Yes, we fully agree with the reviewer’s expert comment that the poisoning effects of cations and anions should be discussed clearly in the manuscript. In fact, when we were doing this work, the poisoning effect has been considered and evaluated based on the investigation of numerous relevant literature. We think that the poisoning effects of cations and anions are unlikely to exist and thereby lead to the sluggish ORR kinetics in HClO₄ electrolyte, which is the target system in this work. The specific reasons are as follows.

First, about the anion poisoning, there indeed exist some previous reports having investigated the poisoning effects of acid electrolyte anions on ORR for both metal and M-N-C catalysts (*ACS Catal.* 2022, 12, 12786–12799; *Commun. Chem.* 2022, 5, 20; *ChemElectroChem* 2021, 8, 2467-2478; *ACS Catal.* 2018, 8, 7104–7112; *J. Phys. Chem. C* 2011, 115, 16087-16097). However, it should be noted that all related experiments in these works were performed in H₂SO₄, HNO₃, H₃PO₄, HCl and HBr electrolytes. It is well documented that the SO₄²⁻, HPO₄²⁻, NO₃⁻, Cl⁻ and Br⁻ anions possess stronger specifically adsorption abilities on surface metal sites (*ACS Catal.* 2022, 12, 12786–12799; *Commun. Chem.* 2022, 5, 20; *ChemElectroChem* 2021, 8, 2467-2478), thus causing the anion poisoning effect. By contrast, the ClO₄⁻ in HClO₄ electrolyte possesses the weakest binding ability on the electrode surface and thus is a non-specifically adsorbed anion, which implies that there exists no anion poisoning in HClO₄ electrolyte. Consequently, the anion poisoning effects in H₂SO₄, HNO₃, H₃PO₄, HCl and HBr electrolytes resulted in worse ORR activity than that in HClO₄ electrolyte (*ACS Catal.* 2022, 12, 12786–12799; *Commun. Chem.* 2022, 5, 20; *ChemElectroChem* 2021, 8, 2467-2478), which is easy to understand. However, it should be highlighted that although there exists no anion poisoning in HClO₄ electrolyte, the M-N-C catalysts still exhibit dramatic activity drop for ORR comparing to the KOH electrolyte (*J. Am. Chem. Soc.* 2013, 135, 15443-15449; *Adv. Funct. Mater.* 2020, 31, 2007423; et al.). In our work, the physical origin is revealed to be the disparate double-layer microenvironments induced by surface charge density difference (Figure R8).

Figure R8. Schematic diagram of the relationship between the disparate double-layer microenvironments and the dramatic active gap under acid and alkaline ORR conditions.

On the other hand, we believe that there also exist no cation poisoning effect in acid media. According to our simulated acid interface on FeCo-N₆-C, it is noted that the electrode surface is much positively charged (Figure 1b in the manuscript), which implies that there exist no H₃O⁺ cations in the double layer region due to the strong electrostatic repulsion. In addition, it should be noted that all interfacial water molecules around M-N center are orderly arranged in the O-down configurations, which suggests that it is also almost impossible for the interfacial water molecules to transfer proton to M-N center to poison it. Therefore, it can be believed that the poisoning effect of cations in acid media is unlikely, which agrees well with Jaouen's results (*J. Phys. Chem. C* 2011, 115, 16087-16097).

We have added the above discussions in the revised manuscript, and the revisions are attached below:

In Lines 351-357 Page 19: "In addition, such sluggish PCET kinetics at acid interface brought about by the positively charged electrode surface and ordered O-down water orientation also means that the protonation of FeCo-N₆ sites is obviously hard. Combined with the nonspecific adsorption property of perchlorate in HClO₄ electrolyte, the poisoning effects of anion and cation seems unlikely to exist to be responsible for the blocked ORR activity in acid, which further implies the key role of disparate double-layer microenvironments in the pH effect of ORR on M-N-C catalysts."

Thank the reviewer again for the expert and valuable comments.

Reviewer #3 (Remarks to the Author):

In this work, the authors investigated the pH-dependence of ORR activity on FeCo-N₆-C DAC by combining the AIMD simulation and slow-growth sampling approach, and demonstrated that the fundamental importance of interfacial double-layer microenvironment is vital when evaluating the reaction mechanism and kinetics. This work provides a new perspective addressing the pH-dependence of ORR activity; however, more detailed and solid demonstration should be given. Here are some comments.

1. Potential of zero charge (PZC) is a fundamental concept when talking about the structure of electric double layer, and the deviations from the PZC in alkaline and acidic media should be evaluated based on the correct determination of PZC. The authors should calculate the PZC of FeCo-N₅-C DAC based on AIMD simulations and compare it with the experimental value.

Response 1: We thank the reviewer very much for raising this important issue.

We apologize for forgetting to provide the PZC value while showing the representative snapshot of FeCo-N₆-C/H₂O system in Figure S1 of the Supplementary Information. The PZC is calculated as -0.24 V vs SHE. Unfortunately, to our best knowledge, it seems that there are still no experimentally measured PZC values for M-N-C catalysts.

Recently, Chan and Liu have calculated the PZC of Fe-N₄-C and Co-N₄-C SACs, which are -0.86 V and -0.42 V vs SHE, respectively (*J. Am. Chem. Soc.* 2021, 143, 9423-9428; *ACS Catal.* 2020, 10, 7826-7835). In this regard, we have also simulated the Fe-N₄-C/H₂O and Co-N₄-C/H₂O interfaces and obtained their PZCs as contrasts, to evaluate the accuracy of our calculation methods for the PZC values of M-N-C systems. The representative snapshots of Fe-N₄-C/H₂O and Co-N₄-C/H₂O interfaces are shown in Figure R9, and their PZCs are calculated as -0.71 V and -0.34 V vs SHE, respectively, which are similar to the values reported by Chan and Liu. This confirms that our calculation method for the PZCs of M-N-C catalysts are accurate and reliable. Therefore, the PZC of FeCo-N₆-C can be determined as -0.24 V vs SHE based on our AIMD simulations. Such a PZC value is fairly lower than the ORR reaction potential, implying that the FeCo-N₆-C electrode possesses much higher positive charge density in acid than that in alkaline, which is fully consistent with the conclusions in the manuscript.

Figure R9. Representative snapshots of (a) Fe-N₄-C/H₂O and (b) Co-N₄-C/H₂O interfaces.

We have added Figure R9 as Supplementary Figure 4a,b in the revised Supplementary Information. Meanwhile, the corresponding discussions have been added as the note of Supplementary Figure 1 and 4, and the revisions are attached below:

In Page S2 of Supplementary Information: “The PZFC of FeCo-N₆-C/water interface is calculated as -0.24 V vs SHE. Unfortunately, to our best knowledge, it seems that there are still no experimentally measured PZFC values for M-N-C catalysts. To evaluate the accuracy of our calculated PZFC values for M-N-C systems, we have also simulated the Fe-N₄-C/water and Co-N₄-C/water interfaces and obtained their PZFCs as contrasts. The representative snapshots of Fe-N₄-C/water and Co-N₄-C/water interfaces are shown in Supplementary Fig. 4a,b, and their PZFCs are calculated as -0.71 V and -0.34 V vs SHE, respectively, which are similar to the values reported by Chan and Liu^{1,2}. This confirms that our calculated PZFCs for M-N-C catalysts are accurate and reliable. Therefore, the PZFC of FeCo-N₆-C can be determined as -0.24 V vs SHE based on our AIMD simulations.”

In Page S5 of Supplementary Information: “The PZFCs for Fe-N₄-C and Co-N₄-C electrodes (Supplementary Fig. 4a,b) are calculated as -0.71 V and -0.34 V vs SHE, which are similar to the values calculated by Chan and Liu^{1,2}. The PZTCs for Fe-N₄-C and Co-N₄-C electrodes with *O₂ (Supplementary Fig. 4c,d) are calculated as 0.17 V and 0.37 V vs SHE.”

2. About the identity of the cations, why was K used to construct the alkaline system, and what about Na or Cs?

Response 2: Thank the reviewer for this insightful comment.

In our work, the main reason for choosing K⁺ to construct the alkaline system is that the 0.1

M KOH aqueous is almost the most common electrolyte for experimental ORR activity measurements of M-N-C catalysts (*Adv. Funct. Mater.* 2020, 2007423; etc.). Of course, it is also possible to choose Na⁺ and Cs⁺, and it can be expected that the identity of cation will not affect the conclusions in the manuscript, because the cation does not approach the electrode surface and change the interfacial microenvironment on an almost uncharged electrode under alkaline condition.

3. About the spectator species. As many papers (10.1021/jacs.9b07712, 10.1021/jacs.2c08743, 10.1021/acscatal.2c00771, 10.1039/d1ta07791k) reported that, the spectator species in axially coordinated manner could affect the reaction energetics substantially, would the axially coordinated species like *O or *OH change the main conclusion?

Response 3: Thank the reviewer very much for bring this important issue to our attention, which has inspired us to deeply consider the effect of axially coordinated species on the ORR reaction energetics and interfacial microenvironment.

Firstly, it is worth emphasizing again that, as shown in Figure S11 of the Supplementary Information, the exact surface oxygenated spectators at alkaline and acid interfaces have been identified as *OH and *O species, respectively, due to that the FeCo-N₆-C electrode is almost uncharged in alkaline while much positively charged in acid. Therefore, the axially coordinated species at alkaline and acid interfaces are also considered as the *OH and *O, respectively. As shown in Figure R10, in the presence of axial oxygenated spectators (*OH_{Ax-OS} and *O_{Ax-OS}), the O₂ molecule adsorbs on the Fe-Co bridge site at alkaline interface while on the Fe top site with end-on configuration at acid interface. Such difference can be attributed to the strong interaction between the *O_{Ax-OS} and the Fe-Co bridge site at acid interface, which severely weakens the interaction strength between the O atom of *O₂ and Co atom. In this scenario, there are still one K⁺ cation and one OH⁻ anion being introduced into the water film to simulate the alkaline environment, and one H₃O⁺ cation and two F⁻ anions being introduced into the water film to simulate the acid environment. The electrode potentials are calculated as ~1.05 V and ~0.90 V for alkaline and acid systems, respectively, which also correspond to the experimental ORR potentials.

Figure R10. Representative snapshots of the interfacial structures at ORR potentials in (a) alkaline and (b) acid electrolytes with the existence of adsorbed O_2 molecules and axial oxygenated spectators (shortened to Ax-OS) on FeCo-N₆-C.

Then, the double-layer structures at alkaline and acid interfaces are analyzed when the axial oxygenated spectators ($*OH_{Ax-OS}$ and $*O_{Ax-OS}$) are considered. As shown in Figure R11, the alkaline and acid interfaces still exhibit the same difference in the double-layer microenvironment features, compared to that without the oxygenated spectators and with the adjacent oxygenated spectators (Figure 2 in the manuscript and Figure S12 in the Supplementary Information). Namely, despite the presence of $*OH_{Ax-OS}$ and $*O_{Ax-OS}$, the interfacial water molecules are still orientated orderly in the form of O-down configuration due to the positively charged electrode surface in acid media, while disorderly in alkaline media (Figure R11a). Correspondingly, the O atom of adsorbed O_2 can form the hydrogen bonds with the H atoms of interfacial water in alkaline media (Figure R11b). Furthermore, it can be seen that the O-O bond length of adsorbed O_2 molecule at alkaline interface is obviously larger than that at acid interface (1.41 Å vs 1.27 Å), due to the assistance of hydrogen bonds formed with interfacial water (Figure R11c,d).

Figure R11. (a) Distribution profiles of water dipole orientations along the surface normal direction at acid and alkaline interfaces when the $^*OH_{Ax-OS}$ and $^*O_{Ax-OS}$ are considered. The insets show that α is defined as the angle between the vector of water dipole (\hat{d}) and the surface normal (\hat{n}). (b) Radial distribution functions between the O atom of adsorbed O_2 molecule that points to the solution and the H atoms of interfacial water. (c) The extraction of O-O bond length of the adsorbed O_2 molecule during the 10 ps AIMD product simulations for alkaline and acid systems. (d) Statistical distributions of the O-O bond length of adsorbed O_2 molecule at acid and alkaline interfaces.

Finally, the ORR mechanisms on FeCo-N₆-C in the presence of axial oxygenated spectators are determined (Figure R12a,b), and the corresponding interface structures of various reaction intermediate states for alkaline and acid systems are shown in Figure R12c,d. It is noted that the ORR proceed via the dissociative and associative mechanisms at alkaline and acid interfaces, respectively. Moreover, at each ORR reaction intermediate state, the hydrogen bonds between the O atoms of surface reaction intermediates and the H atoms of interfacial water can be observed clearly at alkaline interface, which is unlikely at acid interface (Figure R12c,d). Remarkably, Figure R12e shows that the free energy diagrams still show that the PDS at alkaline interface is still much more uphill thermodynamically than that at acid interface (0.96

V vs 0.86 V). Compared to the results at the interfaces without oxygenated spectators (1.43 V vs 1.16 V, Figure 4d in the manuscript) and with adjacent oxygenated spectators (1.39 V vs 0.47 V, Figure S14e in the Supplementary Information), it is seen that although the spectator species in axially coordinated manner can affect the reaction energetics substantially, the conclusions in the manuscript maintain well, namely the dramatic activity gap between alkaline and acid ORR cannot be deciphered from the perspective of the energetics of multistep reaction pathways, but is a kinetic effect.

Figure R12. (a,b) Schematic diagrams of ORR processes in (a) alkaline and (b) acid medias when the $*OH_{Ax-OS}$ and $*O_{Ax-OS}$ (marked by yellow color) are considered. The H_2O marked by red color represent the water molecules which locate in the solution and provide proton to surface oxygenated intermediates, while the H_2O marked by green color mean the reaction

products. (c,d) Representative snapshots of the interfacial structures along the ORR processes in (c) alkaline and (d) acid medias. (e) Free energy diagrams for ORR at alkaline and acid interfaces for $U = 1.0$ V vs RHE.

We have cited these wonderful works recommended by reviewer, and added the above figures in the revised manuscript and Supplementary Information (Figure R10 as Figure 5c,d, Figures R11 and R12 as Supplementary Figures 13 and 15, respectively). Meanwhile, the corresponding descriptions have been added in the revised manuscript, and the revisions are attached below:

In Lines 289-323 Pages 17-18: "To further examine the accuracy and universality of the above results, we have also considered the existence of oxygenated spectators on FeCo-N₆-C catalyst, including the adjacent oxygenated spectators (shortened to Ad-OS)^{34,35} and the axial oxygenated spectators (shortened to Ax-OS)^{14,15,36-38}. The exact Ad-OS at alkaline and acid interfaces are identified as *OH_{Ad-OS} and *O_{Ad-OS} species occupying the Fe-Co bridge sites, respectively (Supplementary Fig. 11). Figure 5a,b show the FeCo-N₆-C/electrolyte interfaces with Ad-OS under alkaline and acid ORR conditions, in which the reactant O₂ molecules are adsorbed on the Fe top sites in the end-on configuration, due to the occupation of Fe-Co bridge sites by the Ad-OS and the stronger affinity of Fe element for oxygenated species. Figure 5c,d show the FeCo-N₆-C/electrolyte interfaces with Ax-OS and *O₂, in which the Ax-OS species under alkaline and acid ORR conditions are also considered as *OH and *O, respectively. It is worth mentioning that despite the presence of Ad-OS or Ax-OS, one K⁺ cation and one OH⁻ anion also need to be introduced to reach the alkaline ORR potentials (~1.00 V and ~1.05 V, respectively), and one H₃O⁺ cation and two F⁻ anions are required to attain the acid ORR potentials (~0.80 V and ~0.90 V, respectively), which means that the electrode surfaces are still almost uncharged in alkaline while positively charged in acid. Correspondingly, the alkaline and acid interfaces still exhibit the same difference in the double-layer microenvironment features (Supplementary Figs. 12 and 13), compared to that without the oxygenated spectators (Fig. 2). Supplementary Figs. 14 and 15 show the determined ORR mechanisms in the presence of oxygenated spectators and the corresponding interface structures of various reaction intermediate states. It is noted that at both alkaline and acid interfaces with the Ad-OS, ORR

proceed via the associative mechanism involving *OOH species, due to the destruction of synergistic effect between adjacent metal active sites by the spectators and the resulting end-on adsorption of O_2 molecule on Fe top site; while in the presence of Ax-OS, the alkaline and acid ORR proceed via the dissociative and associative mechanisms, respectively. Moreover, at each ORR reaction intermediate state, the hydrogen bonds between the O atoms of surface reaction intermediates and the H atoms of interfacial water can be observed clearly at alkaline interface, which is unlikely at acid interface (Supplementary Figs. 14c,d and 15c,d). Remarkably, the ORR free energy diagrams (Supplementary Figs. 14e and 15e) show that although the Ad-OS and Ax-OS species can affect the reaction energetics substantially, the PDS at alkaline interface is still much more uphill thermodynamically than that at acid interface (Fig. 5e), which confirms that the huge ORR activity gap between alkaline and acid media is not attributed to the difference in reaction energetics. All the above results laterally hint that the unfavorable interface double-layer microenvironment and the resultant sluggish PCET kinetics are very likely to be responsible for the inferior ORR performance in acid electrolytes.”

References: “14. Liu, T., Wang, Y. & Li, Y. How pH affects the oxygen reduction reactivity of Fe-N-C materials. *ACS Catal.* **13**, 1717-1725 (2023).

15. Hu, X. et al. What is the real origin of the activity of Fe-N-C electrocatalysts in the O_2 reduction reaction? Critical roles of coordinating pyrrolic N and axially adsorbing species. *J. Am. Chem. Soc.* **144**, 18144-18152 (2022).

36. Wang, Y., Tang, Y. J. & Zhou, K. Self-adjusting activity induced by intrinsic reaction intermediate in Fe-N-C single-atom catalysts. *J. Am. Chem. Soc.* **141**, 14115-14119 (2021).

37. Wei, J. et al. Probing the oxygen reduction reaction intermediates and dynamic active site structures of molecular and pyrolyzed Fe-N-C electrocatalysts by in situ Raman spectroscopy. *ACS Catal.* **12**, 7811-7820 (2022).

38. Hu, X. et al. Understanding the role of axial O in CO_2 electroreduction on NiN_4 single-atom catalysts via simulations in realistic electrochemical environment. *J. Mater. Chem. A* **9**, 23515-23521 (2021).”

4. More calculation details should be given.

(1) Regarding the slow-growth calculation, the potential of mean force profile should be given.

The value of the error bars in Fig. 4 should be given.

(2) Regarding the free energy diagram calculation, what is the time scale or the number of configurations considered to calculate the statistically averaged internal energies of each reaction intermediate? What if the automatic hydrogenation of reaction intermediates in alkaline media?

Response 4: Thank the reviewer very much for these expert suggestions. We are very sorry for not clarifying this important information. The following is our responses point by point.

(1) As shown in Figure R13, the typical potential of mean force profiles derived from the slow-growth simulations for the first PCET reactions at alkaline and acid interfaces and the corresponding free energy profiles along the collective variable (CV, ξ) have been provided. It can be seen that the free energy profiles present a gradual upward trend rather than a parabolic-like shape (*J. Am. Chem. Soc.* 2021, 143, 9423–9428; *J. Chem. Phys.* 2022, 156, 104701). It may be due to the fact that, the slow-growth simulation was performed under constant charge condition, and thus the charge density on FeCo-N₆-C electrode varies greatly, when interfacial water molecule donates a hydrogen atom to the oxygen-containing reaction intermediate and generates a OH⁻ anion, especially around the transition state (TS). As shown in Figure R13b, after the constant potential correction as described in Supplementary Note 1 of the Supplementary Information, the typical parabolic free energy profile (red dashed curve) of an elementary reaction process can be well obtained, and the CV of TS is determined as ~1.15 Å. Such value has also been used to determine the TS in the slow-growth simulation at acid interface (Figure R13d). Note that even within a fairly large CV range around 1.15 Å at acid interface, the intermediate state is always much higher in free energy than that of TS for alkaline interface, which unequivocally demonstrates that the PECT reaction at acid interface is very sluggish.

In addition, the average free energy barriers (ΔG^\ddagger) and the corresponding error bars in Fig. 6 of the manuscript have been listed in Table R1 below.

Figure R13. (a,b) Potential of mean force profiles derived from the slow-growth simulations for the first PCET reaction at alkaline and acid interfaces, respectively. (c,d) The free energy profiles (yellow solid curves) along the collective variable (CV, ξ) at alkaline and acid interfaces, respectively. The red dashed curve in (c) means the free energy profile after constant potential correction.

Table R1. Average free energy barriers (ΔG^\ddagger) and error bars for various slow-growth simulations of the first PCET reactions at alkaline and acid interfaces.

Simulation ID	ΔG^\ddagger (eV)	Error bar (eV)
Alkaline-1	0.328	0.022
Alkaline-2	0.286	0.027
Acid-1	1.318	0.014
Acid-2	1.344	0.019
Acid-3	2.149	0.059

We have added Figure R13 as Supplementary Figure 17 and the above discussions as its note, and added Table R1 as Supplementary Table 1 in the revised Supplementary Information. Meanwhile, the corresponding descriptions have been added in the revised manuscript, and the revisions are attached below:

In Lines 333-335 Page 18: “Supplementary Figure 17 shows the typical potential of mean force profiles derived from the slow-growth simulations for the first PCET reactions at alkaline and acid interfaces and the corresponding free energy profiles along the CV.”

In Line 342 Page 19: “Fig. 6d and Supplementary Table 1 demonstrate that the kinetic free energy barriers for the first PCET reaction in acid ORR are much higher than that in alkaline ORR”

(2) For the free energy diagram calculation, all the statistically averaged internal energies of various reaction intermediates are obtained from the whole 10 ps AIMD production process, which contains 10,000 configurations. In addition, there are no automatic hydrogenation occurring for various reaction intermediates in alkaline media throughout the whole AIMD simulations.

5. Some errors or typos.

(1) The schematic diagram in step 2 is wrong in supplementary Fig. 4.

(2) H₂O in R-1 was missing in supplementary Fig. 5(b) and supplementary Fig. 6(c).

(3) The elementary reaction equations in supplementary Fig. 6(c) are not consistent with the reaction configurations in Fig. 3(b), supplementary Fig. 6(a) and (b).

(4) The second IS should be changed to FS in S18 in SI.

(5) There is a grammatical problem in the title. It might be “...in Alkaline and Acidic Electrolytes”.

Response 5: We thank the reviewer for the careful reading and valuable reminder. The following are our corrections point by point.

(1) We have corrected the schematic diagram of step 2 in Supplementary Fig. 7.

(2) We have added H₂O in the R-1 reaction equations of Supplementary Fig. 8b and Supplementary Fig. 9c.

(3) We have corrected all elementary reaction equations in Supplementary Fig. 9c according to the reaction configurations in Fig. 4b and Supplementary Fig. 9a,b.

(4) We have corrected the “IS” to “FS” in Page S25.

(5) We have corrected the title into “Decisive Role of Disparate Double-Layer Microenvironments Induced by Surface Charge Density Difference in the pH effect of ORR on Metal-Nitrogen-Carbon Catalysts”.

REVIEWER COMMENTS

Reviewer #1 (Remarks to the Author):

The authors have clarified most of the previous concerns. But when it comes to my concerns about the calculated electrode potential, I'm still not convinced by their claim. The authors note that the calculated potential differs by approximately 0.2–0.3 V with and without reaction intermediates. Can the authors estimate the site density of the active site? I suspect that the surface density of Fe-Co sites in the model may be much higher than the actual situation. Then, in an average sense, the surface dipole potential due to the presence of intermediate species would be way overestimated. Also, what is the estimated turn over frequency (TOF) value? TOF values of ORR reactions for Fe-N-C catalysts typically range from 10^{-2} to 10^0 e sites⁻¹ s⁻¹ [JACS Au 2021, 1, 5, 586]. This means that an ORR occurs every 1 to 100 seconds at each active site, meaning the active sites are likely to be mostly empty.

Reviewer #2 (Remarks to the Author):

This is the second revision of the work by Chen et al. The work is very interesting and new, but as stated during the first revision, experimental proofs are missing. For example, the authors are providing theoretical calculation for an Fe-Co catalyst as a case where the activity is much higher in alkaline environment than acidic conditions. Then the authors should provide calculations also for the other case where the activity is higher in acid than in alkaline, i.e. For the industrial and experimental-standard Pt catalyst. This experiment could be considered as blank experiment and could provide a baseline for the following results. The authors are now providing SEIRAS experiments conducted at the catalyst-modified electrode but again a blank experiment is missing. What happens at unmodified electrodes. How would the spectra look like if Pt were used instead of the catalyst. The herein proposed problem is mainly an electrochemical problem. But the authors are not performing electrochemical experiments. A “cheap” electrochemical impedance experiment would have provided much information. Even after recommendations, the authors are not probing the anion poisoning effect. But in the literature the anion poisoning effect in acidic conditions has been proved as in <https://pubs.acs.org/doi/abs/10.1021/acscatal.2c03298>. Considering the high impact factor and the high standards of the Nature Communications Journal, considering the lack of proof and the length of the manuscript which is not easy to read, it is recommended to reject the work.

Reviewer #3 (Remarks to the Author):

The authors have addressed the comments well, and the revised version is acceptable for publication.

Responses to Reviewers' Comments

We sincerely appreciate the reviewers for the expert and valued comments and constructive suggestions which greatly help us to improve the quality of this submission. In the following, we carefully respond the reviewers' comments point by point (texts in blue), and describe the corresponding revisions made on the manuscript and Supplementary Information (orange font). The figures in this Response Letter are named as Figure R**.

Reviewer #1 (Remarks to the Author):

The authors have clarified most of the previous concerns. But when it comes to my concern about the calculated electrode potential, I'm still not convinced by their claim. The authors note that the calculated potential differs by approximately 0.2-0.3 V with and without reaction intermediates. Can the authors estimate the site density of the active site? I suspect that the surface density of Fe-Co sites in the model may be much higher than the actual situation. Then, in an average sense, the surface dipole potential due to the presence of intermediate species would be way overestimated. Also, what is the estimated turn over frequency (TOF) value? TOF values of ORR reactions for Fe-N-C catalysts typically range from 10^{-2} to 10^0 e sites⁻¹ s⁻¹ [JACS Au 2021, 1, 5, 586]. This means that an ORR occurs every 1 to 100 seconds at each active site, meaning the active sites are likely to be mostly empty.

Response: Thank the reviewer very much for these expert and valuable comments. The following is our response point by point.

(1) For the comment "evaluate the active site density in the model and its influence on the potential calculation"

We greatly appreciate the reviewer for bringing this important issue to our attention, namely the consideration of the active site density (SD) in the model. Such issue is indeed often ignored in most of the current ab initio simulation studies, due to the limitation of computational cost when the SD in model is similar to experimental values. However, evaluating the active site density in the model and its influence on the potential calculation and current conclusions is fairly significant.

Inspired by this comment, we have estimated the SD in the synthesized FeCo-N₆-C catalyst

and performed additional AIMD simulations for the interfaces with SD closer to the experimental value. Firstly, according to the composition measurement of the as-prepared FeCo-N₆-C sample by ICP-AES test (*J. Am. Chem. Soc.* 2017, 139, 17281-17284), the experimental SD is estimated as $\sim 1.1 \times 10^{20}$ site/g, which agrees well with other reported values from 3×10^{19} site/g to 2.8×10^{20} site/g (*Nat. Catal.* 2022, 5, 311-323; *ACS Catal.* 2019, 9, 4841-4852; *ACS Catal.* 2018, 8, 1640-1647; *Nat. Commun.* 2015, 6, 8618; et al.). By contrast, the SD of the AIMD model in the manuscript (copied as Figure R1a), which is calculated as 9.2×10^{20} site/g, is indeed obviously higher than the experimental SD of prepared FeCo-N₆-C catalyst.

To evaluate the influence of the SD in interface model on the potential calculation and conclusions in the manuscript, we have established two larger FeCo-N₆-C models (Figure R1b and R1c), which possess the SD of 4.6×10^{20} site/g and 2.3×10^{20} site/g, respectively. Such SD in the enlarged model is fairly closer to the experimental values. For these three models with different SDs (Figure R1a-f), we firstly calculated the work functions (Φ) in vacuum. Figure R1g shows that, the Φ values hardly change for clean FeCo-N₆-C slabs while slightly decrease for O₂ adsorbed FeCo-N₆-C with the SD.

Figure R1. (a-c) Clean and (d-f) O₂ adsorbed FeCo-N₆-C slab models with different SDs in vacuum. (a,d)

9.2×10²⁰ site/g, (b,e) 4.6×10²⁰ site/g and (c,f) 2.3×10²⁰ site/g. (g) The calculated work functions (Φ).

We then performed AIMD simulations for these two enlarged FeCo-N₆-C/water interfaces without and with the O₂ adsorption (Figure R2a-d), and meanwhile their potentials of zero charge (PZCs) were obtained. As shown in Figure R2e, for the clean FeCo-N₆-C/water interfaces, the potential of zero free charge (PZFC) exhibits obvious decrease with the SD decreasing, but not monotonically. By contrast, the potential of zero total charge (PZTC) of O₂ adsorbed FeCo-N₆-C/water interface decreases monotonically with the SD decreasing. Such change trends of the PZTC and PZFC for FeCo-N₆-C/water interfaces as well as the Φ for the O₂ adsorbed FeCo-N₆-C indicate that the SD in model indeed affects the magnitude of the surface dipole potential induced by intermediate and thus the calculated value of the electrode potential.

However, it should be noted that with the SD decreasing from 9.2×10²⁰ site/g to 2.3×10²⁰ site/g which is similar to the experimental SD, the decrease in PZTC and PZFC of FeCo-N₆-C/water interface does not change the significant difference of surface charge density on FeCo-N₆-C electrode in alkaline and acid medias, and thereby the conclusions in the manuscript still maintain well for the model with experimental SD. Specifically, as shown in Figure R2f, when the SD decreases, the PZFC (horizontal dashed line) and the PZTC (horizontal solid line) of FeCo-N₆-C will gradually shift downward, consequently leading to the further deviation (marked by red arrows) of them relative to the ORR reaction potential at both acid and alkaline pHs. Therefore, the electrode will possess much higher positively surface charge density in acidic electrolyte compared to that in alkaline electrolyte, which indicates that the water molecules at acid interface will be much more orderly and orientally-rigid, thereby resulting in much higher barriers of the proton-coupled electron transfer (PCET) steps in ORR process. Such conclusions maintain well at different SDs.

In addition, as shown in Figure 2e, it can be noted that for the FeCo-N₆-C/water models with different SDs, the distinct potential differences between the interfaces with and without reaction intermediates always exist. Therefore, it can be inferred that the intermediate induced potential difference does not seem to be related to SD, although the SD in model is already close to the experimental value.

Figure R2. (a,b) Representative snapshots of FeCo-N₆-C/water interfaces with SD of 4.6 × 10²⁰ site/g and 2.3 × 10²⁰ site/g, respectively. (c,d) Corresponding snapshots of FeCo-N₆-C/water interfaces with O₂ adsorption. (e) Comparison of PZC vs SHE for the clean (PZFC) and O₂ adsorbed (PZTC) FeCo-N₆-C/water interfaces with different SDs. (f) Pourbaix diagram showing the pH dependence of the ORR reaction potential (1.0 V vs RHE is used here), the PZFCs and PZTCs for FeCo-N₆-C/H₂O systems with SD of 9.2 × 10²⁰ site/g (horizontal dashed lines) and 2.3 × 10²⁰ site/g (horizontal solid lines).

We have added the above Figures R1 and R2 as Supplementary Figure 4 and Supplementary Figure 5 in the revised Supplementary Information. Meanwhile, the corresponding descriptions have been added in the revised manuscript, and the revisions are attached below:

In Lines 169-192 Pages 9-10: “Furthermore, the influence of active site density (SD) of FeCo-N₆-C catalyst on the double-layer microenvironments at alkaline and acid ORR interfaces is evaluated, because the SD of the model shown in Fig. 1 (9.2 × 10²⁰ site/g) is much higher than the experimental SD value of the as-prepared FeCo-N₆-C sample (~1.1 × 10²⁰ site/g)²⁰. As shown in Supplementary Figs. 4 and 5, two enlarged FeCo-N₆-C/water interface models, which possess the SD of 4.6 × 10²⁰ site/g and 2.3 × 10²⁰ site/g, respectively, are established and simulated to obtain the potentials of zero charge (PZC). Supplementary

Figure 5e shows that for the clean FeCo-N₆-C/water interfaces, the potential of zero free charge (PZFC) exhibits obvious decrease with the SD decreasing, but not monotonically. By contrast, the potential of zero total charge (PZTC) of O₂ adsorbed FeCo-N₆-C/water interface decreases monotonically with the SD decreasing. Such change trends of the PZTC and PZFC for FeCo-N₆-C/water interfaces indicate that the SD in model indeed affects the magnitude of the surface dipole potential induced by intermediate and thus the calculated values of the electrode potentials. However, it should be noted that with the SD decreasing from 9.2×10²⁰ site/g to 2.3×10²⁰ site/g which is similar to the experimental SD, the decrease in PZTC and PZFC of FeCo-N₆-C/water interface does not change the significant difference of surface charge density on FeCo-N₆-C electrode in alkaline and acid medias, and thereby the significant difference of the double-layer microenvironments at alkaline and acid ORR interfaces. Specifically, as shown in Supplementary Fig. 5f, when the SD decreases, the PZFC (horizontal dashed line) and the PZTC (horizontal solid line) of FeCo-N₆-C will gradually shift downward, consequently leading to the further deviation (marked by red arrows) of them relative to the ORR reaction potential at both acid and alkaline pHs. Therefore, the electrode will always possess much higher positively surface charge density in acidic electrolyte compared to that in alkaline electrolyte, indicating that the water molecules at acid interface will be much more orderly and orientally-rigid.”

(2) For the comment “what is the estimated turn over frequency (TOF) value”

To estimate the TOF value based on our AIMD simulations for the model with a SD of 9.2×10²⁰ site/g (namely the model displayed in the manuscript), we have performed additional slow-growth calculations to obtain the barrier of the rate-determining step (RDS, V→I in Fig. 4a of the manuscript) at alkaline interface. Figure R3 shows the typical interface structures of initial state, transition state and final state along the slow-growth simulation, and the free energy barrier was calculated as ~0.612±0.006 eV. Based on the Arrhenius equation, assuming the pre-exponential factor to be 10⁹-10¹² s⁻¹ (*Chem. Rev.* 1992, 92, 463-480; *J. Catal.* 2020, 388, 30-37; *Electrochim. Acta* 2017, 255, 391-395; *Angew. Chem. Int. Ed.* 2018, 57, 7948-7956; et al.), the rate constant was calculated as ~4.9×10⁻²-4.9×10¹ site⁻¹ s⁻¹. Given the ORR is 4e⁻ process, the TOF was thus estimated to be ~2.0×10⁻¹-2.0×10² e⁻ site⁻¹ s⁻¹,

which agrees with the experimental TOF values (10^{-2} - 10^0 e⁻ site⁻¹ s⁻¹) for ORR on M-N-C catalysts (*JACS Au* 2021, 1, 586-597; *ACS Catal.* 2019, 9, 4841-4852; *Nat. Commun.* 2015, 6, 8618; et al.). This indicates that the calculated results and conclusions based on the model with an SD of 9.2×10^{20} site/g are reasonable and reliable. Certainly, it has to be admitted that, if based on the model with a SD (2.3×10^{20} site/g) that is similar to the experimental SD value, the estimated TOF will be further closer to the experimental TOF, because the surface charge density on FeCo-N₆-C electrode and the interfacial electric field are enhanced with the decrease of SD and PZFC/PZTC, thereby leading to a higher free energy barrier and lower TOF.

Based on the above results, as considered by the reviewer, such a low TOF means that the ORR does not occur on Fe-Co active sites for the majority of the time. However, it should be noted that this does not imply that the active sites are empty for most of the time. Because due to the limitation of reaction energy barrier, the electrode surface should remain in the state before RDS. Based on the ORR pathway and free energy diagrams shown in Figure 4 of the manuscript, it can be inferred that, the electrode surface primarily resides in state IV (namely *O) in acidic media while in state V (namely *OH) in alkaline media.

Figure R3. The interface structures (upper panel) and corresponding close-up (lower panel) of the initial state (IS), transition state (TS) and final state (FS) in the slow-growth simulation for the RDS of alkaline ORR.

In summary, all the above results seem to indicate that our current treatment of electrode

potential calculations and conclusions in the manuscript are reasonable. At the same time, we also revealed that the model with SD that is consistent with the experimental value indeed can give more accurate reaction parameters, but it will not change the conclusions in the current manuscript, and such computational cost will also become very expensive.

Such test about the SD and TOF are very meaningful, which is indeed very important to our research and even related fields. We thank the reviewer very much again for bringing these two points to our attention.

Reviewer #2 (Remarks to the Author):

This is the second revision of the work by Chen et al. The work is very interesting and new, but as stated during the first revision, experimental proofs are missing. For example, the authors are providing theoretical calculation for an Fe-Co catalyst as a case where the activity is much higher in alkaline environment than acidic conditions. Then the authors should provide calculations also for the other case where the activity is higher in acid than in alkaline, i.e. For the industrial and experimental-standard Pt catalyst. This experiment could be considered as blank experiment and could provide a baseline for the following results. The authors are now providing SEIRAS experiments conducted at the catalyst-modified electrode but again a blank experiment is missing. What happens at unmodified electrodes. How would the spectra look like if Pt were used instead of the catalyst. The herein proposed problem is mainly an electrochemical problem. But the authors are not performing electrochemical experiments. A “cheap” electrochemical impedance experiment would have provided much information. Even after recommendations, the authors are not probing the anion poisoning effect. But in the literature the anion poisoning effect in acidic conditions has been proved as in <https://pubs.acs.org/doi/abs/10.1021/acscatal.2c03298>.

Considering the high impact factor and the high standards of the Nature Communications Journal, considering the lack of proof and the length of the manuscript which is not easy to read, it is recommended to reject the work.

Response: Thank the reviewer very much for these valuable comments. The following is our response point by point.

(1) For the comment “lacking experimental proofs and blank experiments” and “a cheap

electrochemical impedance experiment would have provided much information”

Inspired by the reviewer’s comments in the last peer review, we have performed in situ SEIRAS to measure the interfacial water orientation information at alkaline and acid interfaces on FeCo-N₆-C catalyst in the revised version. The catalyst samples were synthesized in the group of Prof. Yuen Wu in the University of Science and Technology of China, who and colleagues previously published the synthesis, physical and electrochemical characterizations of this catalyst (*J. Am. Chem. Soc.* 2017, 139, 17281-17284). The spectral results obtained in present study agreed well with our AIMD simulated results, which provided a strong support for the conclusions.

In addition, as recommended by the reviewer, a cheap electrochemical impedance experiment may be able to provide some information. However, it is difficult for us to imagine how the impedance experiment can accurately probe the interface structure. The electrochemical impedance spectroscopy (EIS) is not a molecule-level tool, and the EIS results usually contain rather mixed information and the conclusions rely on equivalent-circuit fittings which are somewhat arbitrary. Currently, numerous reports have proved the advantages and effectiveness of the combination between AIMD simulation and in situ vibration spectroscopy experiment in determining the interfacial structures (*Nature* 2021, 600, 81-85; *Nat. Catal.* 2022, 5, 900-911; *Nat. Mater.* 2019, 18, 697-701; et al).

Finally, the reviewer recommended performing in situ SEIRAS experiments on an Au substrate without catalyst modification as a blank experiment. For this point, we don’t think the SEIRAS experiments on blank Au electrode is necessary and would help. It is well-known that the electrochemical SEIRAS results are usually given in the form of difference spectra by choosing the spectrum measured at a certain potential under the same reaction conditions as the reference spectrum, which can exclude the influence of background and interfering signals and amplifies signal changes caused by the interfacial structural changes. In electrochemical SEIRAS, people don’t use the spectrum on the blank electrode as reference.

(2) For the comment “adding AIMD simulations and in situ SEIRAS for Pt catalyst”

We sincerely thank the reviewer for this comment. However, we don’t think this is useful and necessary for present study. The aim of this paper is to understand the activity difference of

metal-nitrogen-carbon catalysts toward ORR in acid and alkaline media. Currently, it is an important consensus that the ORR activity of metal-nitrogen-carbon catalysts in acidic electrolytes is much lower than that in alkaline. We believe that our results now give a much reasonable explanation on this important problem from the perspective of electric double layer, which is also the first report to the best of our knowledge. For Pt, the story could be very different. To perform AIMD simulations for Pt catalyst requires very costive simulations on different single crystal planes of Pt. Besides, in situ SEIRAS study of ORR on Pt is currently very challenging. A combined AIMD simulations and SEIRAS study of ORR on Pt could give another highly important paper. The results, however, should not directly benefit the results and conclusions of the present paper.

(3) For the comment “evaluating the anion poisoning effect”

Inspired by the reviewer’s comments in the last peer review, we have added detailed discussions about the anion poisoning effect, with combined literature research on this aspect and our simulation results. We have highlighted that the goal of our study is to explain the origin of the experimentally observed difference in ORR activity of M-N-C catalysts in KOH and HClO₄ solutions (Figure R4a). Due to that the ClO₄⁻ is a non-specifically adsorbed anion, it can be believed that there does not exist the anion poisoning effect in HClO₄ solution. In previous reports that focus on the pH effect of ORR on M-N-C catalysts in KOH and HClO₄ solutions (*J. Am. Chem. Soc.* 2022, 144, 18144-18152; *ACS Catal.* 2023, 13, 1717-1725; et al.), no ClO₄⁻ anion poisoning effect has been observed. As for the so-called anion poisoning effect based on the reference (*ACS Catal.* 2022, 12, 12786-12799) mentioned by the Reviewer, it should be noted that all the used acid electrolytes contain the specifically adsorbed anions (e.g., Cl⁻, HSO₄⁻ and CH₃CO₂⁻), thus resulting in the decrease in ORR activity on M-N-C catalyst (Figure R4b,c). In addition, Jaramillo et al. have also reported that, the ORR activities in the acid electrolytes containing specifically adsorbed anions (e.g., HNO₃, H₂SO₄, H₃PO₄, HCl and HBr) were much lower than that in HClO₄ solution, let alone when comparing to the alkaline ORR activity (Figure R5, from *Commun. Chem.* 2022, 5, 20 and *ChemElectroChem* 2021, 8, 2467-2478). It is easy to understand that, the further decrease of ORR activity in the acid electrolytes containing specifically adsorbed anions, as compared with that in HClO₄ solutions,

is due to the anion poisoning effect. However, the aim of present paper is to unveil the origin of the pH effect rather than the anion effect. Our results show that the interfacial water structures could be a reason. We don't think the anion poisoning effect applies to our study.

Figure R4. (a) Schematic diagram of the relationship between the disparate double-layer microenvironments and the dramatic active gap under acid and alkaline ORR conditions. (b) Polarization curves at 1600 rpm, and 5 mV s⁻¹ in O₂-saturated electrolytes toward the ORR in various electrolytes. (c) TOF determined at 0.6 V vs RHE for the ORR in various electrolytes.

Figure R5. (a) Representative third-cycle-averaged N₂-subtracted oxygen reduction reaction (ORR) rotating disk electrode (RDE) cyclic voltammogram (CV) of the Pt disk in pH 1 HClO₄, HNO₃, and H₂SO₄. (b) Representative RDE/RRDE (1600 rpm), N₂-CV-subtracted, ORR cyclic voltammograms (20 mV s⁻¹) of Ag thin films as a function of acid electrolyte (pH 1).

Reviewer #3 (Remarks to the Author):

The authors have addressed the comments well, and the revised version is acceptable for publication.

Response: We sincerely thank the reviewer for the support and recognition of our work.

REVIEWER COMMENTS

Reviewer #1 (Remarks to the Author):

The authors have done a great job addressing all the previous comments. The new calculation results have clarified my previous concern about the calculation of the electrode potential. As such I agree to the publication of this manuscript.

Reviewer #2 (Remarks to the Author):

This is the third revision for this work. My concerns were not satisfied by the answers of the authors, not during the first revision, not during the second revision. The authors provided further results which contain 100% error. The work will not be of relevant significance to the field. The data do not support conclusions and claims. Many flaws are present in the data analysis. The methodology is not sounding. It was asked to the authors to combine experimental results to theoretical ones but the authors were not able to follow the recommendations and also for the last set of provided data only theoretical results were provided.

Responses to Reviewers' Comments

We sincerely appreciate the reviewers for the valued comments and constructive suggestions. In the following, the reviewers' comments are addressed point by point (texts in blue), and the corresponding revisions made on the manuscript are also described (orange font).

Reviewer #1 (Remarks to the Author):

The authors have done a great job addressing all the previous comments. The new calculation results have clarified my previous concern about the calculation of the electrode potential. As such I agree to the publication of this manuscript.

Response: We sincerely thank the reviewer for the positive evaluation of our work.

Reviewer #2 (Remarks to the Author):

This is the third revision for this work. My concerns were not satisfied by the answers of the authors, not during the first revision, not during the second revision. The authors provided further results which contain 100% error. The work will not be of relevant significance to the field. The data do not support conclusions and claims. Many flaws are present in the data analysis. The methodology is not sounding. It was asked to the authors to combine experimental results to theoretical ones but the authors were not able to follow the recommendations and also for the last set of provided data only theoretical results were provided.

Response: We thank the reviewer very much for these valuable comments. The followings are our point-by-point responses.

(1) For the comments "The authors provided further results which contain 100% error" and "Many flaws are present in the data analysis"

We don't understand exactly why the reviewer have these comments. It is apparently untrue that the results provided in the revision contain 100% error, since the other reviewers have accepted them well. In addition, the reviewer didn't specifically point out what flaws are there in the revision. We feel that we have carefully responded all confusions of the reviewer, and the results and data analysis in our manuscript are accurate and believable.

(2) For the comment “The work will not be of relevant significance to the field.”

As stated in the manuscript, the present work possesses both the fundamental and technological significance. Fundamentally, our work provides a unique molecular-level insight into the pH effect of ORR activity on M-N-C catalysts from the perspective of electrocatalytic interface structures, which should be of fundamental importance for electrochemistry. In the technological aspect, it is hoped that our work would open new and feasible avenues to improve the activity of M-N-C catalysts for proton exchange membrane fuel cells (PEMFCs) through the structural and functional modulation of electrocatalytic interface. Besides, the findings in our work should explain that why the recent results on improving the ORR activity and electrochemical stability of M-N-C catalysts in acid by the introduction of protic ionic liquids (ILs) (*ACS Energy Lett.* 2019, 4, 2104-2110; *ACS Appl. Mater. Interfaces* 2019, 11, 11298-11305; et al.).

(3) For the comments “The methodology is not sounding”, “The data do not support conclusions and claims” and “It was asked to the authors to combine experimental results to theoretical ones but the authors were not able to follow the recommendations and also for the last set of provided data only theoretical results were provided”

Firstly, it is worth highlighting that the in-situ surface-enhanced infrared absorption spectroscopy (SEIRAS) measurements have been conducted to verify the results of AIMD simulations on the different water orientations at alkaline and acid interfaces on FeCo-N₆-C catalyst, in responding to the reviewers' comments. The SEIRAS is a widely-used and accurate methodology in determining the interfacial structures of electrocatalytic systems, especially for the interfacial water structure (*Nature* 2021, 600, 81-85; *Nat. Catal.* 2022, 5, 900-911; *Nat. Mater.* 2019, 18, 697-701; et al). The spectral results obtained in present study agreed well with the AIMD simulated results, which provided a strong support for the conclusions. Meanwhile, it indicates that the methodology in our work is reasonable and reliable.

In this manuscript, we have performed standard AIMD simulations and SEIRAS experiments to meticulously obtain the interfacial structures, the ORR pathway and the reaction energetics of various elementary steps. Besides, we have also performed slow-growth simulations to

obtain the kinetic barriers of proton-coupled electron transfer (PCET) steps. The theoretical and experimental results and implications have been well correlated, providing a solid support for the conclusions.

In light of the reviewer's comments and the editors' suggestions, we have made further revisions in the manuscript to avoid possible confusion and misunderstanding.

(1) We have weakened some claims which seem to be lacking direct experimental evidence.

- In this work, we mainly attributed the ORR activity difference of M-N-C catalysts in acid and alkaline electrolytes to the disparate interfacial double-layer structures which are pH-dependent. As for the poisons of anion and cation in acid, we have pointed out that they seem to play a minor role according to our simulated results. However, we have weakened the statement for this point. Specifically, the statement "such sluggish PCET kinetics at acid interface brought about by the positively charged electrode surface and ordered O-down water orientation also means that the protonation of FeCo-N₆ sites is obviously hard. Combined with the nonspecific adsorption property of perchlorate in HClO₄ electrolyte, the poisoning effects of anion and cation seems unlikely to exist to be responsible for the blocked ORR activity in acid" has been revised to "the scarcity of hydronium and the fairly high PCET barrier at acid interface caused by the much positively charged electrode surface and ordered O-down water orientation imply that the protonation of FeCo-N₆ active center may be unlikely. Combined with the nonspecific adsorption property of perchlorate in HClO₄ electrolyte which is the system targeted in this study, the poisoning effects of anion and proton seems to play a rather minor role in inhibiting the ORR activity" (Lines 383-388).
- In addition, about the effects of anions in acid media on ORR, we have added a description, namely "However, further studies are also required to holistically explore the possible effects of anions in acid electrolytes on ORR activity^{42,43}." (Lines 390-391). The related references have also been cited as refs. 42 and 43 (Lines 595-599).
- About the experimental verification for the simulated results, the reviewer suggested that the surface charge density and interfacial water orientation could be measured. Due to the huge challenge of measuring the surface charge density and its correlation with the water

orientation, we have performed the in situ SEIRAS to verify the simulated interfacial water structures in acid and alkaline medias. However, it is undeniable that if the surface charge density of M-N-C materials under ORR conditions can be obtained, this will provide additional solid support for the conclusions in this work. We have added this important point in the manuscript, namely “Certainly, apart from the characterization of the interfacial water structure by in situ spectroscopy, the direct measurement and/or indirect evaluation of surface charge density are also worth exploring to provide additional experimental verification for the simulated results, but this is still challenging so far” (Lines 251-254).

(2) we have added some discussions on the limitation of several results as well as the difficulties in modeling the material in real experimental conditions in the main text of manuscript.

- Due to the limitation of computational cost, the active site density (SD) in model is hard to be set to approach the experimental values. This limitation has been added in the manuscript, namely “It is noted that the SD in such model is much higher than that of the experimentally synthesized FeCo-N₆-C sample ($\sim 1.1 \times 10^{20}$ site/g)²⁰, because the computational cost will be daunting if the SD in model is set to similar to experimental value” (Lines 440-443).
- Based on the model with higher SD, it can be expected that the calculated reaction barriers would not be exactly same with the actual values in experimental conditions. However, we have found that the SD would not change the differences of surface charge densities and interfacial microenvironments between alkaline and acid systems, as well as the relative sizes of the reaction barriers of PCET steps. This limitation has been added in the manuscript, namely “Although the more actual barriers can be obtained when using the interface model with a SD closer to the experimental value, it has been stated that the SD would not change the differences of surface charge densities and interfacial microenvironments between alkaline and acid systems, as well as the relative sizes of the barriers of PCET steps” (Lines 372-375).

(3) we have also described some potential future directions that could help to strengthen the current study.

- According to our study, optimizing the interfacial water orientation and constructing the

hydrogen bonds with surface oxygenated intermediates should receive much attention to further accelerate the acid ORR kinetics on M-N-C catalysts. The specific future direction has been described in the manuscript, namely “highlight immense opportunities to improve the ORR kinetics in acid environment through the structural and functional modulations of electrocatalytic interface. In this regard, the introduction of protic ionic liquids and organic molecules and/or dispersing the charge accumulation on metal center through electrode structure design may be effective practice strategies to optimize the interfacial water orientation and construct the hydrogen bonds with surface oxygenated intermediates” (Lines 378-383).

- For modeling the M-N-C material in real experimental conditions, it is suggested that the machine-learning potential based molecular dynamics simulations would be useful, but it is still fairly challenging currently. This point has also been described as a future direction in the manuscript, namely “In this regard, the machine-learning potential based molecular dynamics simulations would play an important role in modeling the M-N-C material in real electrocatalytic conditions, although it is still fairly challenging currently^{51,52}” (Lines 443-445). The related works have also been cited as Refs. 51 and 52 (Lines 616-621).